# Evolutionary Alpha Factor Discovery with Large Language Models for Sparse Portfolio Optimization

## Abstract

Sparse portfolio optimization is a fundamental yet challenging task in quantitative finance, as traditional approaches that rely on historical return estimates and static objectives often struggle to adapt to shifting market dynamics. To address this, we propose a new framework that leverages large language models (LLMs) to automate the discovery and iterative refinement of alpha factors tailored for sparse portfolio construction. By reformulating asset selection as a top-m ranking problem guided by factor signals, our framework integrates an evolutionary feedback loop to continuously enhance the factor pool based on performance. Extensive experiments across five Fama–French benchmark datasets and two real-world datasets (US and China) show that our approach consistently outperforms both statistical and optimization-based baselines, particularly in high-volatility and large-universe settings. Ablation studies further highlight the importance of prompt design, factor diversity, and the choice of LLM backend. These results suggest that language-model-guided factor generation offers a promising, interpretable, and adaptive solution for portfolio optimization under sparsity constraints.

## 1 Introduction

Sparse portfolio optimization aims to construct a portfolio by selecting at most $m$ assets from a universe of $n$ candidates ($m \ll n$) to optimize key performance metrics, such as cumulative return, risk, or risk-adjusted return. Due to the combinatorial nature of the selection constraint ($\ell_0$-norm) and the non-convexity of most financial objectives, the problem is known to be NP-hard (Lin et al., 2024a) and lacks efficient closed-form solutions.

To tackle this impoartant yet challenging problem, classical approaches utilize greedy selection, convex relaxation, mixed-integer programming, and sparsity-regularized optimization models (Brodie et al., 2009; Lai et al., 2018; Dai & Wen, 2018; Kremer et al., 2020; Gunjan & Bhattacharyya, 2023; Lin et al., 2024a) to compute the exact solutions or their approximations under simplifying assumptions. However, they suffer from two critical limitations: (1) the generated investment suggestions lack interpretability and are less understandable to general public; (2) the algorithms are often sensitive to the hyperparameter choices, leading to unstable performance across market regimes.

To enhance interpretability and adaptability, recent strategies (Ang, 2014; Fan et al., 2016) have adopted factor-based portfolio construction, where we search factors to map an asset's historical features, such as prices, returns, or volatility, into a score indicating its relative attractiveness. These factors guiding asset ranking and investment allocation are usually called *alpha factors* and widely employed in both academia and industry. Despite more transparent, such approaches pose new challenges: identifying effective factors often requires domain expertise, manual tuning, and frequent re-validation. Moreover, many factors have poor transferability across different market conditions and usually degrade quickly in live markets. To address this, recent studies have explored machine learning techniques for alpha factor discovery (Zhang et al., 2020; Yu et al., 2023a), but the vast search space of asset combinations limits the scalability and effectiveness of these methods. In addition, as shown in Figure 1, many factor libraries crafted by existing methods (e.g., from Qlib (Yang et al., 2020)) suffer from *sparse decay*, a phenomenon where sharp performance drops in sparse regimes (e.g., selecting top-10 assets). Sparse decay issue makes it challenging to apply ex-

isting methods in sparse portfolio optimization and indicates that many factors crafted by them are not precise enough to identify the *very best assets* under the sparse constraint. In this context, we need more expressive and adaptive factor generation frameworks to address the concerns above from multiple aspects.

On the other hand, large language models (LLMs) have shown impressive capabilities in financial applications, including forecasting (Yu et al., 2023c;b) and multimodal market analysis (Bhatia et al., 2024; Yu et al., 2024). These works demonstrate that LLMs can effectively model complex patterns in financial data. However, most LLM-based applications focus narrowly on predictive tasks Yu et al. (2023c); Nie et al. (2024);

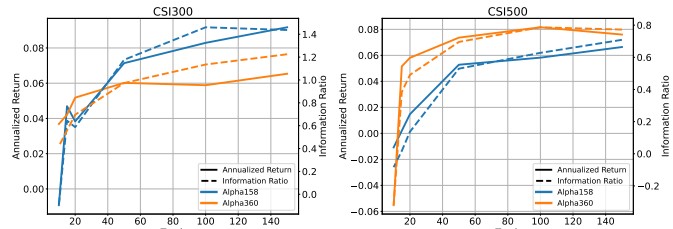

Figure 1: Sparse decay: performance of Alpha158 and Alpha360 factors on CSI300 and CSI500 (2022–2025). Solid lines: annualized returns; dashed lines: information ratios.

Zhao et al. (2024), such as price movement classification or sentiment analysis, without addressing the downstream challenges of portfolio construction. Given LLMs' generative nature and ability to synthesize new patterns, recent studies have begun exploring their use in alpha factor discovery (Wang et al., 2023; Yuan et al., 2024; Wang et al., 2024; Li et al., 2024; Shi et al., 2025a;b; Tang et al., 2025). While they show the potential of LLMs for generating investment factors, these approaches have two key limitations. First, they rely heavily on human guidance, and treat factor mining as a static, one-shot process. This neglects the dynamic nature of financial markets, especially in the case of long-term investment where alpha signals often decay. Second, studies frequently test these factors on large portfolios of 50 or more assets. This approach overlooks the practical constraints of real-world portfolio management, where factors like implementation costs, risk control, and the need for interpretability necessitate focusing on a much smaller, sparse set of assets.

To address the limitations of existing alpha mining algorithms and the challenges of sparse portfolio optimization, we propose a novel LLM-driven framework that dynamically generates, evolves, and constructs portfolio under sparse constraints. Our main contributions are: (1) We introduce a dynamic LLM-based alpha factor mining framework and are the first to incorporate evolutionary algorithms into LLM-driven factor discovery, enabling continual factor generation and refinement; (2) We design an end-to-end pipeline that unifies factor mining with sparse portfolio optimization, seamlessly linking factor generation, evaluation, and asset selection. This avoids the need for complex model training, reduces implementation overhead, and achieves state-of-the-art performance across multiple datasets. Our framework adapts by analyzing real-time market performance and proposing new factors, providing a powerful tool for navigating the evolving market environment; (3) We provide an in-depth empirical analysis of LLM behaviors in financial factor discovery and optimization tasks, examining their creativity, diversity, and stability. This sheds light on both their strengths in generating effective signals and their limitations under dynamic market conditions.

## 2 RELATED WORKS

**Sparse Portfolio Optimization.** Sparse portfolio optimization has been extensively explored through a range of methodological innovations designed to balance return, risk and sparsity. One line of work incorporates $\ell_0$-regularization into the Markowitz framework (Witt & Dobbins, 1979) to encourage sparsity and enhance out-of-sample stability (Brodie et al., 2009; Fastrich et al., 2015). By contrast, another line of works focuses on index tracking under strict $\ell_0$-constraints, offering methods that provide explicit control over asset selection and tracking error (Li et al., 2022). Furthermore, some other approaches (Lai et al., 2018) leverage optimization techniques such as ADMM to solve short-term sparse portfolio problems, particularly relevant in high-frequency trading contexts. To promote diversity, some studies introduce structured sparsity through grouped penalties like the SLOPE regularization (Kremer et al., 2020). More recent works (Lin et al., 2024b) propose unified frameworks using indicator relaxations and proximal algorithms for optimizing sparse mean-CVaR portfolios. In addition, efficient global solvers have been developed for maximizing Sharpe ratios under cardinality constraints (Lin et al., 2024a).

**Alpha Factors Mining.** Alpha factor mining evolves dramatically with the advancement of machine learning. Early approaches employ classical machine learning approaches such as evolutionary algorithms (Zhang et al., 2020) and reinforcement learning (RL) (Yu et al., 2023a; Zhu & Zhu, 2025) to generate and refine alpha factors. These approaches are summarized in a theoretical framework in (Shi et al., 2025a). However, these traditional methods lack operational diversity and suffer from engineering complexity, restricting their ability to generate expressive and adaptive factors.

Recent works leverage large language models (LLMs) to enhance factor mining. The Alpha-GPT series (Wang et al., 2023; Yuan et al., 2024) pioneered human-AI collaborative factor generation, though it still relies on manual feedback and lacks full autonomy. Further studies employ financial signals as guidance to polish LLM-generated factors (Wang et al., 2024) or use symbolic experience chains to improve their interpretability (Li et al., 2024). The latest work apply Monte-Carlo trees for more effective factor searching (Shi et al., 2025b) and chain-based searching (Cao et al., 2025). While these methods demonstrate the potential of LLMs in factor mining, they primarily operate in a static manner, failing to account for the dynamic nature of financial markets.

**Automated Algorithm Design Driven by LLMs.** The factor mining task shares key similarities with automated algorithm design: both require exploring complex search spaces to discover high-performing and interpretable solutions. Recent advancements have leveraged LLMs to automate the design of algorithms and heuristics, particularly for combinatorial optimization. For example, the Evolution of Heuristics (EoH) framework (Liu et al., 2024) combines LLMs with evolutionary computation to generate optimization heuristics autonomously. Extending this, the MEoH framework (Yao et al., 2025) introduces multi-objective optimization to balance solution quality and efficiency using a dominance-dissimilarity mechanism for population management. Further innovations like ReEvo (Ye et al., 2024) integrate reflective evolution, where LLMs iteratively critique and refine solutions to enhance reasoning. These frameworks demonstrate the promising potential of LLMs to automate the creation of adaptive, high-performance heuristics, which is a capability equally critical for dynamic factor mining in finance.

# 3 PRELIMINARIES

## 3.1 PORTFOLIO OPTIMIZATION UNDER THE $\ell_0$ NORM CONSTRAINTS.

Portfolio optimization aims to determine the allocation of capital across $n$ assets to maximize investment performance while controlling risks. The sparse portfolio problem under the $\ell_0$-norm constraint seeks portfolio weights $\boldsymbol{w} \in \mathbb{R}^n$ that optimize a given objective $g(\boldsymbol{w})$ such as investment returns, subject to budget, non-negativity and sparsity constraints:

$$\text{maximize}_{\boldsymbol{w}} \quad g(\boldsymbol{w}) \text{ subject to } \boldsymbol{w}^\top \mathbf{1} = 1, \ \boldsymbol{w} \geq \mathbf{0}, \ \|\boldsymbol{w}\|_0 \leq m, \tag{1}$$

where $\|\boldsymbol{w}\|_0$ denotes the $\ell_0$-norm of $\boldsymbol{w}$, i.e. the number of nonzero entries in $\boldsymbol{w}$, and thus $m$ specifies the maximum number of assets selected in the portfolio ($m \ll n$). Without the loss of generality, we assume unit total investment and let the non-negative vector $\boldsymbol{w}$ in a simplex. In a sequential decision making process of $T$ steps, we may adjust the portfolio weights $\boldsymbol{w}$ to approximate the optimality of Problem (1) in each time stamp $t = 0, \ldots, T$ and use $r_{t+1}$ to represent the corresponding realized total return. In this context, we use $P_t = \prod_{s=1}^{t} r_s$ to represent the *cumulative asset value* in each time stamp and consider key performance metrics as below:

- **Cumulative Wealth (CW)** is the total portfolio return ratio, i.e., $\text{CW} = P_T/P_0$. This metric is particularly useful for examining long-only sparse selection strategies that do not involve leverage, i.e., a significant gain suggests the ability of consistently picking top winners.

- **Sharpe Ratio (SR)** is defined as the difference between the returns of the investment and the risk-free return, divided by the standard deviation of the investment returns. Specifically, given a series of realized portfolio returns $\{r_t\}_{t=0}^{T}$ with different time stamps and the risk-free return $r$. Sharpe ratio is computed as $\text{SR} = \frac{\mathbb{E}_t[r_t] - r}{\sqrt{\text{Var}_t[r_t]}}$ where $\mathbb{E}_t[r_t]$ and $\text{Var}_t[r_t]$ calculate the expectation and variance of $r_t$, respectively. From a statistical perspective, the value of SR depends on the length $T$ of the series. To estimate the annual SR, it is standard practice to multiply the monthly SR by $\sqrt{12}$ or multiply the daily SR by $\sqrt{252}$, respectively, in which 12 and 252 represent the approximate number of trading months and trading days in a year. In this work, we assume $r = 0$.

## 3.2 FACTOR SEARCHING

In quantitative finance, an *alpha factor* assigns a score to each asset based on its past features such as price, return, and volatility. Formally, for asset $i = 1, \ldots, n$, its history over a window of length $T$ with $d$ features is represented as a matrix $\boldsymbol{X}_i \in \mathbb{R}^{d \times T}$. An alpha factor $f$ maps this matrix to a scalar score $f(\boldsymbol{X}_i)$, where a higher score suggests the asset is more attractive under a chosen objective. Factors are built from raw features, constants, and operators, including: (1) unary functions (e.g., $\mathrm{abs}(\cdot)$, $\log(\cdot)$); (2) binary operations (e.g. $+$, $-$, $\times$, $/$); and (3) time-series operators (e.g., $\mathrm{Sum}(\mathrm{volume}, 5d)$). Such factors can be viewed as computation trees, with features as leaves and operators as internal nodes. The aim of *factor search* is to find interpretable formulas that generate reliable signals from noisy markets. These signals can rank and select assets directly, or be fed into machine learning models for return prediction. However, using complex models often reduces interpretability and increases overfitting risks (Shi et al., 2025a; Tang et al., 2025). On the other hand, classical approaches like genetic programming are also inefficient, as they explore blindly in the vast space of possible formulas (Tang et al., 2025).

If we use $\{r_t\}_{t=0}^{T}$ and $\{s_t\}_{t=0}^{T}$ to represent the realized returns and factor scores with different time stamps, respectively. Serving as predictive signals, alpha factors are typically evaluated by correlation-based metrics as follows:

- **Information Coefficient (IC)** measures the Pearson correlation between factor scores and next-period returns: $\mathrm{IC} = \mathbb{E}_t[\rho(s_t, r_{(t+1)})]$ where $\rho$ calculates the correlation. IC measures the overall linear predictive power of alpha factors.

- **Rank Information Coefficient (RankIC)** measures the Spearman rank correlation between factor scores and next-period returns: $\mathrm{RankIC} = \mathbb{E}_t[\rho(\mathrm{Rank}(s_t), \mathrm{Rank}(r_{t+1}))]$. RankIC evaluates how well alpha factor rankings match realized return rankings.

- **Information Ratio (IR)** measures the ratio of expectation over the standard derivation for either IC or RankIC overtime. The corresponding metric is denoted as **ICIR** and **RankICIR**, respectively. A higher value indicates more consistent predictive performance.

## 4 METHODOLOGY

### 4.1 OVERVIEW

As illustrated in Figure 2, compared with optimization-based methods, factor-based approaches offer a key advantage: they are extremely fast, as new portfolios can be generated instantly from daily prices without expensive retraining. Large language models (LLMs) further boost this benefit by rapidly producing a large number of candidate factors in a far more efficient way than manual design. Building on this, we propose to employ LLMs as generators to replace the traditional static "feedback–modify" cycle with an evolutionary process, where factors are co-evolved based on their historical performance by backtesting. Specifically, we encode both the task scenario and past factor statistics into LLM instructions, guiding the generation of task-specific alpha factors. The generated alpha factors are then filtered by their performance, with only a small set of the best alpha factors remained. Since we only keep the top-performed alpha factors, we can bypass intermediate predictive models as in traditional methods (Yang et al., 2020; Shi et al., 2025b) and instead directly apply simple aggregation (e.g., equal-weighting) to produce asset-level scores. Portfolios are then constructed by selecting the top-$m$ assets under the $\ell_0$ sparsity constraint.

Overall, our LLM-powered evolutionary alpha factor discovery establishes a closed-loop workflow generating interpretable alpha factors that adapt quickly to market dynamics. Unlike traditional machine learning methods, our framework eliminates the need for extensive training and hyper-parameter tuning and thus boosts the efficiency. In addition, leveraging the strong generalization capabilities of LLMs (), our method mitigates overfitting and demonstrates robust performance across diverse markets in extensive experiments. Finally, concerns about LLM hallucination are naturally addressed by evolutionary algorithm, which filters out incorrect or ineffective alpha factors caused by hallucination during the selection process, ensuring the reliability and effectiveness of the generated alpha factors. The pseudo-code and more details of our method are provided in Appendix A.

As shown in Figure 2, our framework for sparse portfolio optimization has three key modules: (1) **LLM-based Alpha Factor Generator** which guides the generation of task-specific alpha factors; (2) **Evolutionary Search Mechanism** which continuously evaluates, refines, and updates the factor library to make system adapt to shifting market conditions and mitigate the problem of alpha decay; (3) **Factor-to-Portfolio Construction** which aggregates the discovered factors to construct the sparse portfolio. We introduce these modules in the following sections.

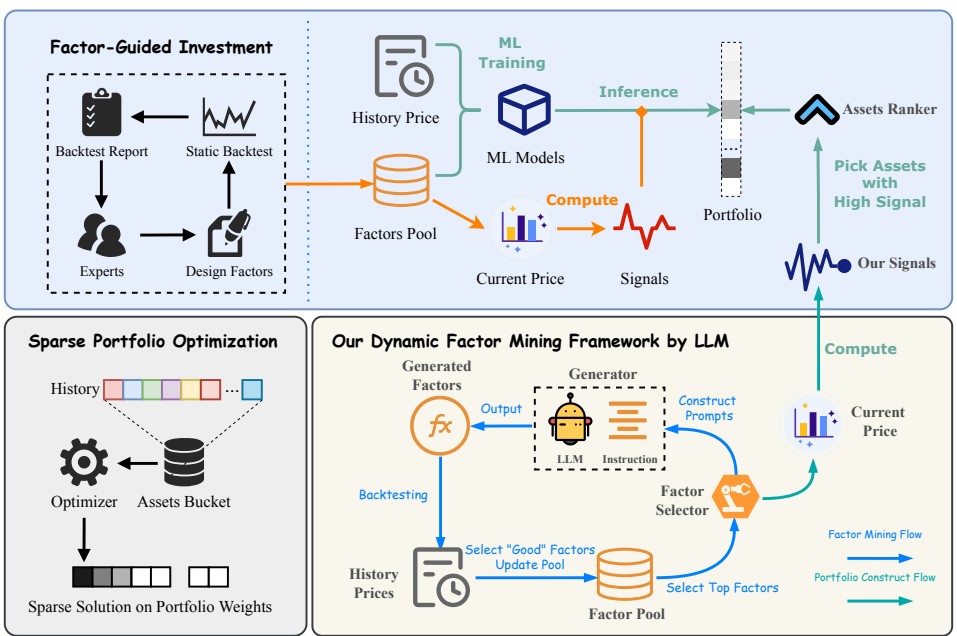

Figure 2: Comparison of portfolio construction paradigms. Optimization-based methods directly solve for sparse weights, and traditional factor-guided investment depends on expert-designed factors with ML aggregation. In contrast, our LLM-driven framework autonomously generates and evolves factors, removing reliance on manual mining and directly guiding portfolio construction.

## 4.2 LLM-BASED ALPHA FACTOR GENERATOR

As shown in Figure 3, our alpha factor generator is an LLM agent responsible to produce alpha factors for further selection and evaluation. We design system prompts with detailed instructions, including background on alpha factors, generation rules, and definitions of EA operations (details in Appendix C) for LLM to generate interpretable alpha factors in a unified format. In addition to the prompt, the LLM agent receives information about the current top factors and their recent backtesting performance as heuristics to generate new alpha factors, then we apply two filtering stages to ensure output quality: (1) **Format check.** We verify executability, operator validity, and syntactic correctness of the generated expressions. Failures often arise from LLM *hallucinations*, such as producing undefined functions, mismatched arguments, or invalid syntax. When such issues occur, a retry mechanism is triggered to regenerate candidates until a sufficient number of valid factors are obtained. (2) **Performance check.** All factors that pass the format stage are instantly backtested. Candidates showing poor or unstable predictive ability are discarded to avoid polluting the factor library with spurious signals.

Through filtering, we ensure that only factors meeting both structural and empirical quality requirements are admitted into the evolving factor library. This design mitigates risks from unreliable LLM outputs and guarantees that subsequent portfolio construction builds upon a robust foundation.

## 4.3 EVOLUTIONARY SEARCH MECHANISM

The core innovation of our framework lies in its dynamic evolutionary search mechanism, which overcomes several limitations of prior approaches: (1) Traditional methods adopt a static, one-shot search, where discovered factors are directly applied over long horizons. This static design fails to

adapt to market regime shifts, leading to rapid factor decay. (2) Static full-scale searches also require a large number of trials before producing usable factors, making them costly and inefficient.

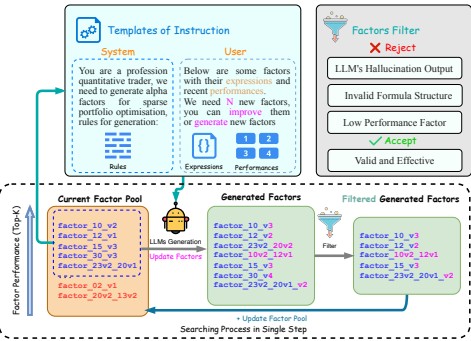

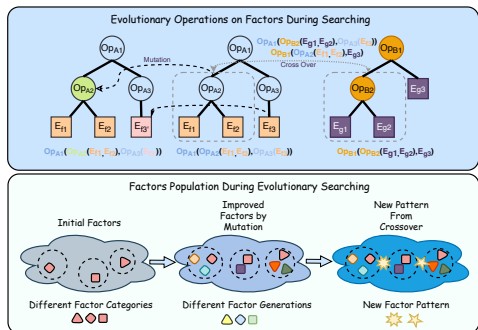

Figure 3: End-to-end alpha factor generation pipeline. Prompts guide LLMs to improve or generate factors, which are then filtered by format and performance checks before updating the factor pool.

Figure 4: Illustration of evolutionary factor search. **Top:** Example AST-based mutation and crossover operations on factors. **Bottom:** Factor population evolves as mutation and crossover introduce diversity.

Our dynamic evolutionary algorithm (EA) resolves these issues by maintaining an incrementally updated factor library throughout the investment process. Unlike chain-based or tree-based search (Cao et al., 2025; Shi et al., 2025b), EA naturally promotes population diversity through mutation and crossover, enabling the library to accumulate a broad set of factors tailored to different market conditions. Moreover, continuous backtesting ensures that factors admitted into new searches are always performance-validated under the most recent regimes. This closed-loop design makes the framework both adaptive and practical in real markets. As illustrated in Figure 4, we visualize EA at two levels: (1) the detailed search process with factor expressions represented as abstract syntax trees (ASTs), where mutation modifies parameters (e.g., operator window lengths) or replaces operators, and crossover exchanges structural components between factors to create new patterns; and (2) the population-level evolution that maintains diversity. Leveraging the generative capability of LLMs, mutation and crossover can be extended beyond basic edits to synthesize novel combinations of weak signals, forming linear or nonlinear composites. Consequently, EA drives sustained factor diversity, supporting long-term portfolio construction with robustness across market environments.

## 4.4 PORTFOLIO OPTIMIZATION UNDER AUTONOMOUS FACTOR SEARCH

Based on the modules of alpha factor generation and evolutionary search mechanism, we can construct the portfolio in an end-to-end manner, consisting of the following three stages. We illustrate the workflow in Figure 5 and provide the pseudo-code in Algorithm 1.

**(1) Factor Library Warmup.** We initialize a seed factor pool from basic technical features (Table 3). Each factor is evaluated over a look-back window to collect performance statistics, such as CW and RankIC. This provides the initial guidance for the evolutionary search and forms the knowledge base for factor selection.

**(2) Periodic Evolutionary Search.** At a fixed search frequency $T$ (e.g., every $T$ trading days), we update the factor pool as introduced in previous sections. The process evaluates the existing pool using the most recent backtesting results, selects the top-$k$ factors, and applies mutation and crossover to generate new candidates. Newly generated factors are filtered by executability and performance, while outdated ones are pruned, ensuring adaptivity and diversity in the pool.

**(3) Daily Portfolio Construction.** On each trading day $t$, every asset $i$ is assigned scores $f_j^i(t)$ from the top-$k$ factors $\{f_j\}_{j=1}^k$ and ranked by the aggregated score $s_i(t) = \frac{1}{k}\sum_{j=1}^k f_j^i(t)$. The top-$m$ assets are selected under the $\ell_0$ sparsity constraint. Unless specified, the portfolio weights $\{w_i(t)\}$ for selected assets $\mathcal{S}_t$ are equal, i.e., $w_i(t) = \frac{1}{m}, \forall i \in \mathcal{S}_t$. In Appendix E.3, we compare equal weighting with score-proportional weighting, in which $w_i(t) = \frac{\max\{s_i(t),0\}}{\sum_{j\in\mathcal{S}_t}\max\{s_j(t),0\}}$

We assume daily rebalancing, where weights are updated at market close and positions are held until the next trading day $t + 1$. This closed-loop workflow ensures that the system continuously adapts to market dynamics while maintaining interpretability and efficiency.

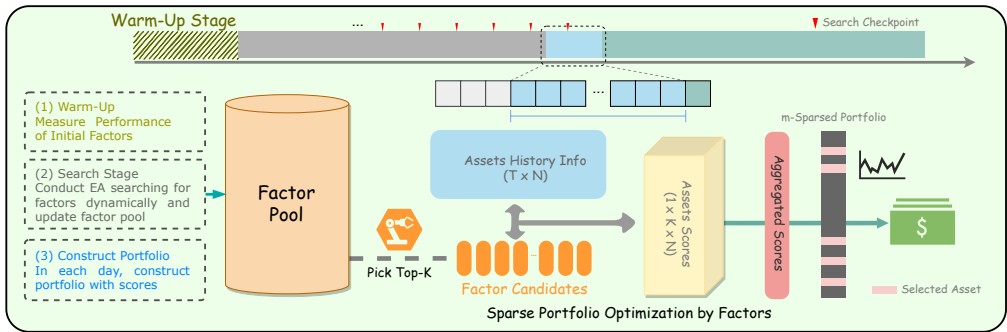

Figure 5: Overall illustration of the portfolio optimization process. The framework begins with a warm-up to evaluate initial factors, followed by dynamic search to evolve the factor pool. At each step, top factors are selected, aggregated into scores, and used to construct sparse portfolios for daily investment decisions.

## 5 EXPERIMENTS

### 5.1 EXPERIMENTAL SETUP

**Datasets.** We evaluate all competing methods on two categories of datasets: academic benchmarks and real-world asset pools. We first use five standard benchmark datasets from Kenneth R. French Data Library (Fama & French, 2023): *FF25*, *FF32*, *FF49*, *FF100* and *FF100MEOP*. We also evaluate our method on two major markets, China and the US, across diverse market regimes. For China, we use CSI300 and CSI500 constituents from 2022.01 to 2025.06, and for the US, we use US50 and NASDAQ100 components from 2019 to 2024. We also construct a data set on the Hong Kong market. More information about dataset is provided in the Appendix B.1.

**Settings.** We benchmark our framework against a broad range of portfolio construction approaches. Baselines are divided into two categories: *non-sparse* strategies, including equal-weighted $(1/N)$ portfolios (as a proxy for market performance), Minimum Conditional Value at Risk (Min-CVaR), and Maximum Sharpe Ratio (Max-Sharpe); and *sparse* strategies, including SSPO (Lai et al., 2018), machine learning selectors such as XGBoost and LightGBM, and advanced methods such as mSSRM-PGA (Lin et al., 2024a) and ASMCVaR (Lin et al., 2024b). For implementation, we employ two widely used online LLM services, GPT-4.1 and DeepSeek-V3, and report results using *Cumulative Wealth (CW)* and *Sharpe Ratio (SR)*. Portfolios are updated on a daily basis, with a new factor search performed every five trading days. Factor scores are normalized to $[-1, 1]$, and at each step the top five factors are selected to construct the portfolio. To reduce search costs, we perform the search in a smaller pool (e.g., CSI300) and apply the factors to both pools (e.g., CSI300 and CSI500) within the same market. Additional details on assumption on metrics, experiment settings and parameters are provided in Appendix D.

### 5.2 RESULTS OF PORTFOLIO PERFORMANCE

We evaluate our approach on both Fama–French benchmark portfolios and real-market datasets. Due to space constraints and our focus on practical applications, we provide performance and a detailed discussion of benchmark results in Table 7 in Appendix E.1.

Table 1 highlights several key findings across the US50, NASDAQ100, CSI300, and CSI500 datasets. Under the 10-asset selection ($m = 10$), our method consistently outperforms both traditional baselines and recent sparse optimization approaches, with particularly striking gains on US50 where CW and SR improve by nearly seven times compared to the equal-weight strategy. On NASDAQ100, the improvements are also substantial, indicating that the method adapts well to markets with higher volatility and different asset structures. For CSI300, which presents the challenge of a

much larger universe, our approach demonstrates strong generalization by achieving peak CW and stable SR values, confirming its scalability. When the portfolio expands to 15 assets ($m = 15$), CW values naturally decline as capital is spread across lower-ranked assets, yet the Sharpe Ratios remain competitive and in most cases superior to existing baselines, showing that our strategy is robust to changes in portfolio size. Another important observation is that the method maintains reliability even under higher sparsity ratios, continuing to deliver consistent improvements across markets. Together, these results provide clear evidence that our approach is both scalable and resilient, achieving strong performance across different asset universes and levels of sparsity. We analyze the effect of transaction costs in Appendix E.4 and confirm that our methods remain robust. The impact of varying the number of factors used in portfolio construction is discussed in Appendix E.5, .

Table 1: Evaluation of Cumulative Wealth (CW↑) and Sharpe Ratio (SR↑) on four real-market datasets (US50, NASDAQ100, CSI300, and CSI500).

| Group | Method | US50 | | NASDAQ100 | | CSI300 | | CSI500 | |
|---|---|---|---|---|---|---|---|---|---|
| | | CW↑ | SR↑ | CW↑ | SR↑ | CW↑ | SR↑ | CW↑ | SR↑ |
| | 1/N | 4.562 | 0.072 | 4.065 | 0.069 | 1.087 | 0.014 | 1.110 | 0.016 |
| Baseline | Min-cVaR | 1.779 | 0.038 | 1.068 | 0.018 | 0.992 | 0.003 | 1.151 | 0.023 |
| | Max-Sharpe | 4.495 | 0.061 | 3.276 | 0.062 | 1.008 | 0.007 | 0.968 | 0.001 |
| | LGBM | 4.182 | 0.063 | 5.054 | 0.071 | 2.334 | 0.072 | 1.560 | 0.038 |
| | XGBoost | 6.313 | 0.077 | 7.372 | 0.072 | 1.420 | 0.032 | 2.309 | 0.057 |
| m=10 | mSSRM-PGA | 5.121 | 0.059 | 9.001 | 0.068 | 0.881 | 0.002 | 0.656 | -0.013 |
| | ASMCVaR | 10.259 | 0.073 | 17.627 | 0.076 | 1.453 | 0.030 | 1.991 | 0.042 |
| | **Ours-DeepSeek** | **25.101** | **0.132** | **37.709** | **0.134** | 3.437 | 0.079 | 4.839 | 0.092 |
| | **Ours-GPT** | 22.905 | 0.130 | 25.896 | 0.126 | **4.962** | **0.098** | **5.502** | **0.098** |
| | LGBM | 3.899 | 0.062 | 4.687 | 0.069 | 1.812 | 0.055 | 1.404 | 0.032 |
| | XGBoost | 5.607 | 0.076 | 5.341 | 0.072 | 1.348 | 0.029 | 1.907 | 0.048 |
| m=15 | mSSRM-PGA | 4.976 | 0.062 | 8.410 | 0.071 | 0.787 | -0.010 | 0.613 | -0.027 |
| | ASMCVaR | 11.124 | 0.074 | 12.345 | 0.068 | 1.658 | 0.035 | 1.603 | 0.023 |
| | **Ours-DeepSeek** | 13.978 | 0.114 | 19.793 | 0.121 | 2.510 | 0.067 | **3.894** | **0.088** |
| | **Ours-GPT** | **14.707** | **0.117** | **21.096** | **0.127** | **3.218** | **0.082** | 3.568 | 0.081 |

## 5.3 ABLATION STUDIES

To analyze the role of key components in our framework, we conduct ablation studies on the US50 and HSI45 dataset (Table 2). All experiments use the DeepSeek backend with three repeated trials, and results are reported as "mean ± standard deviation". We focus on three aspects: (1) the information used in the LLM prompt, including *w/o Sparse Heuristic* (exclude nonsparse backtest, $m = 25$), *w/o Numeric* (exclude numeric metrics), *w/o Quality* (exclude stability and consistency indicators), and *w/o Performance* (exclude RankIC and Recall@N feedback); (2) the initial factor library, where we test *w/o TA* (exclude technical analysis seeds) and *Initial Factor* (handcrafted library only, no LLM factors); (3) LLM-Generator's capacity: the number of generated candidates per step (M).

Table 2: Overall real-market portfolio performance metrics (CW = Cumulative Wealth, SR = Sharpe Ratio, RankIC = Rank Information Coefficient, RankICIR = Information Ratio of RankIC).

| Method | US50 | | | | HSI45 | | | |
|---|---|---|---|---|---|---|---|---|
| | CW↑ | SR↑ | RankIC↑ | RankICIR↑ | CW↑ | SR↑ | RankIC↑ | RankICIR↑ |
| Initial Factor | 6.254 | 0.081 | 0.005 | 0.352 | 1.364 | 0.031 | 0.018 | 0.950 |
| Ours-Deepseek | 32.993±6.044 | 0.149±0.003 | 0.027±0.001 | 1.582±0.050 | 3.193±0.923 | 0.076±0.015 | 0.022±0.001 | 1.412±0.103 |
| w/o Sparse Heuristic | 19.248±3.642 | 0.125±0.003 | 0.020±0.004 | 1.262±0.265 | 2.198±1.094 | 0.051±0.030 | 0.027±0.015 | 1.508±0.828 |
| w/o Numeric | 23.414±3.814 | 0.133±0.007 | 0.023±0.004 | 1.334±0.201 | 2.943±0.516 | 0.072±0.007 | 0.024±0.002 | 1.435±0.118 |
| w/o Quality | 24.152±7.988 | 0.133±0.016 | 0.021±0.003 | 1.178±0.202 | 2.402±0.404 | 0.060±0.010 | 0.016±0.003 | 1.011±0.200 |
| w/o Performance | 9.549±5.869 | 0.094±0.019 | 0.009±0.009 | 0.487±0.467 | 1.168±0.036 | 0.020±0.002 | 0.009±0.002 | 0.522±0.106 |
| w/o TA Factors | 5.367±1.652 | 0.074±0.011 | 0.002±0.004 | 0.117±0.241 | 1.875±1.354 | 0.037±0.037 | 0.008±0.016 | 0.494±0.940 |
| M=5 | 39.126±17.516 | 0.101±0.010 | 0.017±0.002 | 1.036±0.079 | 2.448±0.155 | 0.058±0.006 | 0.024±0.002 | 1.384±0.092 |
| M=15 | 13.246±8.741 | 0.084±0.022 | 0.012±0.005 | 0.701±0.228 | 2.004±0.767 | 0.044±0.022 | 0.026±0.008 | 1.417±0.450 |
| M=20 | 11.952±5.272 | 0.071±0.003 | 0.003±0.000 | 0.187±0.004 | 2.029±0.088 | 0.044±0.002 | 0.021±0.005 | 1.257±0.214 |

Table 2 shows that TA factors and performance feedback are crucial. Removing performance metrics causes the sharpest decline, confirming that backtest-driven feedback is essential for factor evolution. Likewise, excluding TA formulas weakens portfolio stability, underscoring their value as structural priors. For the results of generated candidates per step ($M$), smaller $M$ (e.g., 5) improves average performance but increases variance, while larger $M$ adds diversity yet dilutes quality, reducing

Sharpe Ratio, RankIC, and RankICIR. This degradation stems from (i) limited LLM context/budget producing lower-quality or redundant expressions, and (ii) weaker evolutionary pressure in early stages. Thus, moderate generation sizes strike a better balance between diversity and stability.

We further study the effect of warm-up duration by adjusting the number of pre-backtest search steps. Longer warm-up shows little benefit, suggesting that off-line factor mining alone fails to capture evolving markets. In contrast, our co-evolutionary design, which couples generation with continuous backtesting, adapts more effectively. To validate this, we freeze the factor pool at intermediate checkpoints (record ratio $[0.1, 0.9]$). As shown in Figure 6, both portfolio returns and factor quality improve steadily with more evolution, confirming that ongoing search enhances robustness and predictive power, even under limited LLM query budgets.

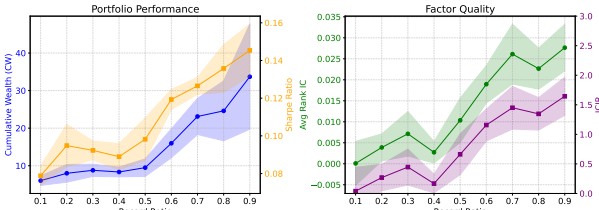

Figure 6: Portfolio performance (US50) under varying record ratios with 5 selected factors. Left: Cumulative wealth (CW) and Sharpe Ratio (SR) with ±1 standard deviation. Right: Factor quality across record ratios: average RankIC (green) and RankICIR (purple).

### 5.4 ANALYSIS AND DISCUSSION

In factor generation, the creativity of LLMs combined with the evolutionary mechanism enables the discovery of diverse new factors, as illustrated in Appendix F.1. A qualitative review of LLM-generated factors (Figure 11) demonstrates that these expressions are both executable and interpretable, often encoding sound trading logic in complex yet readable structures. This process mitigates overfitting risks and preserves diversity within the factor pool. A comprehensive analysis of portfolios constructed by our framework is provided in Appendix F.2. Experimental results show consistent and significant improvements in real-market portfolio performance across multiple datasets. Both cumulative wealth and Sharpe Ratio clearly outperform traditional baselines and recent optimization-based methods under various asset selection settings. These gains stem from our LLM-driven, end-to-end factor generation pipeline. As shown in Figure 15, the selected assets exhibit strong temporal alignment with market regimes: the framework allocates to growth-leading stocks during bull markets and shifts to more stable assets in downturns. Figure 13 further confirms this adaptability, where the method consistently outperforms baselines by limiting drawdowns in bear phases while capturing more upside in rallies. Beyond portfolio-level performance, we also analyze the effect of generated factors. Figure 17 shows that factor scores evolve dynamically across time and markets, indicating responsiveness to shifting financial conditions. Our evidence is consistent with the notion that while asset returns are driven by a small set of factors, the specific factors are not static and the market exhibits high complexity due to their ever-changing nature.

## 6 CONCLUSION, LIMITATION AND FUTURE WORK

This work introduces a novel large language model-guided framework for sparse portfolio optimization under $\ell_0$ constraints. By reframing the asset selection task as a factor-based ranking problem, the method leverages LLMs to autonomously generate, evolve, and refine alpha factors over time. Extensive experiments on both benchmark datasets and real-world markets demonstrate not only consistent gains in cumulative wealth and Sharpe ratio, but also enhanced robustness across different market regimes and asset universes. Beyond raw performance, the approach provides interpretability by producing transparent factor expressions, offering both practical trading signals and insights into underlying market structures. These contributions highlight the potential of LLM-driven factor discovery as a flexible and scalable paradigm for sparse portfolio construction.

There are several limitations remain: The framework relies on few search agents, leading to variability from LLM outputs. Factors are currently implemented only as Python functions, which constrains integration with quantitative platforms. Moreover, the approach uses only historical prices, leaving out richer signals such as news or alternative data. Future work will address these issues through multi-agent search, tighter integration, and incorporation of multimodal information.

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

## REPRODUCIBILITY STATEMENT

To facilitate the reproducibility of our work, we provide the following materials as supplementary resources: sample code, searching checkpoints, and datasets. These materials are made available for reference, enabling other researchers to replicate the key components of our study.

## USAGE OF LLM

In our workflow, large language models (LLMs) are primarily employed in two ways. First, LLMs are used to assist with **data visualization**. By generating clear descriptions, titles, and captions, the models help in producing figures and tables that are both accurate and accessible.

Second, LLMs support **writing refinement and formatting**. They are leveraged to polish drafts, improve sentence flow, and enforce consistency in style and structure. This includes aligning with academic writing conventions, standardizing terminology, and adapting the text to required formatting guidelines.

# A  ALGORITHM DETAILS

## A.1  ALGORITHM OF EVOLUTIONARY FACTOR SEARCHING FRAMEWORK

In this section, we present the detailed procedure of our proposed LLM-Guided Evolutionary Factor Search framework, which integrates dynamic factor generation, population pruning, and backtesting under sparse portfolio constraints. The full workflow is summarized in Algorithm 1.

---

**Algorithm 1** LLM-Guided Evolutionary Factor Search and Sparse Portfolio Optimization

---

1: **Input:** Return matrix $\mathcal{R}$, window $N_t$, search interval $s$, drop threshold $T_{\text{drop}}$, initial library $\mathcal{A}_{\text{init}}$
2: Initialize factor pool $\mathcal{F} \leftarrow \mathcal{A}_{\text{init}}$, current time $t \leftarrow 1$
3: Initialize performance tracker $\mathcal{P}[\alpha]$ for all $\alpha \in \mathcal{F}$
4: *// Warm-up: evaluate initial factor performance*
5: **for** $t = 1$ to $N_t$ **do**
6:     **for** each factor $\alpha \in \mathcal{F}$ **do**
7:         Evaluate score and update $\mathcal{P}[\alpha][t]$
8:     **end for**
9: **end for**
10: Initialize portfolio value $V \leftarrow 1.0$, baseline value $B \leftarrow 1.0$
11: **for** $t = N_t + 1$ to $T$ **do**
12:     **if** $t \bmod s = 0$ **then**
13:         *// Clean factor pool before LLM generation*
14:         $\mathcal{F}, \mathcal{P} \leftarrow \texttt{clean\_factor\_pool}(\mathcal{F}, \mathcal{P}, \text{max\_size}, \text{keep\_top\_n})$
15:         Generate performance report $\mathcal{R}_{\text{perf}} \leftarrow$ recent $\mathcal{P}[\alpha]$ from $t - s$ to $t - 1$
16:         Generate prompt using top-performing factors via $\texttt{filter\_factor\_versions}$ on $\mathcal{R}_{\text{perf}}$
17:         Call LLM with retry:
18:             $\mathcal{A}_{\text{gen}}, \text{success} \leftarrow \texttt{call\_llm\_with\_retry}(\text{prompt}, \text{model})$
19:         **if** success **then**
20:             $\mathcal{F} \leftarrow \mathcal{F} \cup \mathcal{A}_{\text{gen}}$
21:         **else**
22:             Log failure and proceed with current pool $\mathcal{F}$
23:         **end if**
24:     **end if**
25:     *// Evaluate new factors and update performance*
26:     **for** each $\alpha \in \mathcal{F}$ **do**
27:         Compute current score $\alpha_t$ and update $\mathcal{P}[\alpha][t]$
28:         **if** $\alpha$ is newly generated **then**
29:             Validate $\alpha$ and check against benchmark
30:             **if** $\alpha$ fails validation or underperforms benchmark **then**
31:                 $\mathcal{F} \leftarrow \mathcal{F} \setminus \{\alpha\}$
32:             **end if**
33:         **end if**
34:     **end for**
35:     *// Select factor subset and execute backtest*
36:     Select top $k$ valid factors $\mathcal{F}_t \leftarrow \texttt{filter\_factor\_versions}(\mathcal{P})$
37:     Compute asset scores $\mathbf{s}_t$ using $\mathcal{F}_t$
38:     Normalize scores, compute weights $\mathbf{w}_t$, handle NaNs or fallback to equal weight
39:     Compute portfolio return $r_t = \mathbf{w}_t^\top \mathbf{r}_t$
40:     **if** error or invalid return **then**
41:         Set $r_t \leftarrow \bar{r}_t$ (market average)
42:     **end if**
43:     Update portfolio value $V \leftarrow V \cdot r_t$
44:     Update baseline $B \leftarrow B \cdot \bar{r}_t$
45: **end for**
46: **Output:** Final factor pool $\mathcal{F}$, performance metrics (CW, ICIR)

---

To ensure reliable factor generation and effective population management, we incorporate two key mechanisms in our evolutionary framework. First, a retry-based querying process is used to handle the instability and occasional rate limits of online language models. This process repeatedly attempts generation until either a valid response is received or a predefined retry limit is reached. Second, a filtering mechanism validates the generated outputs by checking both their quantity and syntactic correctness against predefined thresholds. Only factors that pass these checks are admitted into the population. Together, these mechanisms provide robustness and continuity for the evolutionary loop, making the framework stable even under real-time or large-scale experimental conditions.

In parallel, a dedicated filtering mechanism is applied to balance performance selection with population diversity. Because each alpha factor can evolve into multiple variants over time, the system retains only two representatives for each base factor: the most recent version and the strongest-performing version, measured by a chosen quality metric such as final portfolio value or average RankIC. This strategy reduces redundancy while ensuring that structurally distinct variants of the same conceptual factor remain in the pool. As a result, the evolving population preserves both quality and variety, which are essential for sustained exploration and adaptive portfolio construction.

---

**Algorithm 2** Aggregate Factor Search Records

---

1: **Input:** List of checkpoint paths $\mathcal{P}$, record limit $N$, ratio limit $r$
2: Load search records $\mathcal{R}_1, \mathcal{R}_2, \ldots, \mathcal{R}_k$ from paths $\mathcal{P}$
3: Determine aligned length $L = \min\{\text{len}(\mathcal{R}_i)\}$
4: **if** $N > 1$ **then**
5:      $L \leftarrow \min(L, N)$
6: **end if**
7: **if** $r < 1$ **then**
8:      $L \leftarrow \min(L, \lfloor L \cdot r \rfloor)$
9: **end if**
10: Truncate all $\mathcal{R}_i$ to length $L$
11: Initialize merged record list $\mathcal{M} \leftarrow []$
12: **for** $t = 1$ to $L$ **do**
13:      Initialize merged performance map $\mathcal{P}_m$, quality map $\mathcal{Q}_m$, expression map $\mathcal{E}_m$
14:      **for** each record list $\mathcal{R}_i$ **do**
15:          Extract record at step $t$: $R_i^t$
16:          Filter portfolio performance and quality using `filter_factor_versions`
17:          **for** each factor $f$ in filtered performance **do**
18:              Get final value $v_f$ and mean RankIC $q_f$
19:              **if** $f$ not in $\mathcal{P}_m$ or $v_f > \mathcal{P}_m[f]$ **then**
20:                  Update $\mathcal{P}_m[f] \leftarrow v_f$, $\mathcal{Q}_m[f] \leftarrow q_f$
21:                  Update $\mathcal{E}_m[f]$ from expression library or record
22:              **else if** $v_f = \mathcal{P}_m[f]$ and $q_f > \mathcal{Q}_m[f]$ **then**
23:                  Update $\mathcal{Q}_m[f] \leftarrow q_f$
24:              **end if**
25:          **end for**
26:      **end for**
27:      Append merged result at step $t$ to $\mathcal{M}$
28: **end for**
29: **return** $\mathcal{M}$

---

To ensure the effectiveness and reliability of LLM-generated factors, we employ a two-stage filtering mechanism. First, we perform an *executability check* to verify that the factor expression is syntactically valid—free of grammatical errors, undefined variables, invalid operators, or mismatched argument structures. Only expressions that successfully pass this structural validation proceed to the second stage, where we evaluate their *output quality*. In this step, each factor is executed on historical test time series to examine its numerical stability and meaningfulness. Factors that produce abnormal values, constant outputs, or non-informative patterns are discarded. This filtering process ensures that only structurally correct and empirically sound factors enter the evolutionary pool.

## A.2 Algorithm of Searching Record Aggregation

To enhance the robustness of our evolutionary factor search, we design a distributed-style parallel searching mechanism, where multiple independent search processes are executed concurrently in Algorithm 2. At each step, we aggregate the factors discovered across these parallel searches by merging their performance records and factor pools. This aggregation not only enlarges the effective factor pool size, allowing for greater diversity and exploration and also helps mitigate the variance that may arise between individual searchers due to stochastic outputs. Importantly, this strategy reduces the impact of service quality fluctuations in online LLMs by smoothing out anomalies or failures in any single search process, thereby yielding more stable and reliable factor evolution results.

## A.3 Factor Library Details

We design our initial factor library based on a set of fundamental and widely adopted price-based operations, ensuring both interpretability and generality. As summarized in Table 3, the library includes basic statistical measures such as mean return, volatility, and momentum, as well as standard technical indicators like moving averages, Bollinger Band width, and RSI. Each factor is constructed using atomic operations on historical price or return sequences, providing a transparent and modular starting point for further factor evolution.

Table 3: Factor Library Details: Type and Mathematical Definitions

| Factor Name | Type | Mathematical Definition | Description |
|---|---|---|---|
| mean_return_w | Basic | $\frac{1}{w}\sum_{i=1}^{w} r_{t-i}$ | Mean of past $w$ returns |
| std_return_w | Basic | $\sqrt{\frac{1}{w}\sum_{i=1}^{w}(r_{t-i}-\bar{r})^2}$ | Standard deviation of past $w$ returns |
| momentum_w | Basic | $\frac{P_t}{P_{t-w}} - 1$ | Price momentum over window $w$ |
| max_drawdown_w | Basic | $\min\left(\frac{P_i-\max_{j\leq i} P_j}{\max_{j\leq i} P_j}\right), i = t-w,\ldots,t$ | Max drawdown in the window |
| sharpe_ratio_w | Basic | $\frac{\bar{r}}{\sigma_r}$ | Sharpe ratio of past $w$ returns |
| volatility_w | Basic | $\text{std}(\log(P_i/P_{i-1})), i = t-w+1,\ldots,t$ | Volatility using log returns |
| price_position_w | Basic | $\frac{P_t-\min(P_{t-w+1:t})}{\max(P_{t-w+1:t})-\min(P_{t-w+1:t})}$ | Price position in range |
| log_return_1 | Basic | $\log\left(\frac{P_t}{P_{t-1}}\right)$ | 1-step log return |
| ma_w | TA | $\frac{1}{w}\sum_{i=0}^{w-1} P_{t-i}$ | Simple moving average |
| bb_width_w | TA | $\frac{2\cdot\sigma_P}{\text{MA}_w}$ | Bollinger Band width (std normalized by MA) |
| ema_ratio_w | TA | $\frac{P_t}{\text{EMA}_w}$ | Ratio of price to EMA |
| rsi_14 | TA | $100 - \frac{100}{1+\frac{\text{Avg Gain}}{\text{Avg Loss}}}$ | Relative Strength Index over 14 days |

The initial factor library is intentionally kept concise for two main reasons. First, the selected factors already cover the fundamental operations commonly used to characterize financial time series, such as measures of return, volatility, momentum, and standard technical indicators. Second, we deliberately start from these basic atomic operations rather than adopting existing complex factor pools (e.g., Alpha101) because many of those factors involve advanced constructs like cross-sectional ranking or multi-asset relationships. At this stage, generating Python functions via LLMs to handle such cross-asset logic remains challenging and does not align with our design goal of producing independent, interpretable evaluation functions for single asset. Moreover, since our framework aims to demonstrate how an LLM can evolve factor expressions from the simplest building blocks, we intentionally avoid including human-engineered factors to ensure that any performance gains reflect the evolution process rather than inherited domain knowledge.

## A.4 Analysis of Complexity

Sparse portfolio optimization under the $\ell_0$-norm is a highly complex problem due to its combinatorial nature, where the goal is to select a limited number of assets from a large universe while optimizing performance metrics. Traditionally, solving this problem requires exhaustive search methods, such as mixed-integer programming or greedy algorithms, which are computationally expensive and sensitive to hyperparameter choices. In contrast, our method leverages large language models

(LLMs) to automate the generation and evolution of alpha factors. This significantly reduces the complexity by transforming the asset selection task into a top-$m$ ranking problem, which is computationally efficient and can be handled directly by LLMs. Moreover, the use of LLMs allows for dynamic factor generation without the need for retraining, making the process training-free and adaptable to evolving market conditions. This efficient, prompt-guided approach reduces the need for large-scale, iterative optimization and enhances both scalability and adaptability. As a result, our method offers a more efficient and flexible solution compared to traditional $\ell_0$-norm optimization techniques, while maintaining the ability to quickly adjust to market dynamics.

## B  DATASET DETAILS

To eliminate scale differences across stocks and ensure compatibility with existing optimization-based sparse portfolio methods, we first compute the *daily relative returns* as:

$$r_t = \frac{p_{\text{close},t}}{p_{\text{close},t-1}} \tag{2}$$

These returns are used as input features for factor evaluation and portfolio construction. To reconstruct a normalized price series, we set the initial price of all assets to 100 and iteratively apply the relative returns:

$$p_t = 100 \times \prod_{i=1}^{t} r_i \tag{3}$$

This normalization ensures that all assets begin from a common scale, facilitating fair comparison across the asset universe and aligning with the assumptions of most sparse portfolio optimization frameworks.

### B.1  BENCHMARK DATASET

We use five widely-adopted benchmark datasets: FF25, FF32, FF49, FF100, and FF100MEOP, from the Kenneth R. French Data Library, following the setting in Lin et al. (2024a;b), the number in each dataset's name (e.g., 25) refers to the number of constituent portfolios. Each dataset contains monthly price-relative sequences, where each

Table 4: Information of used real-world monthly benchmark datasets.

| Data Set | Region | Time | Months | # Assets |
|---|---|---|---|---|
| FF25 | US | Jul/1971–May/2023 | 623 | 25 |
| FF32 | US | Jul/1971–May/2023 | 623 | 32 |
| FF49 | US | Jul/1971–May/2023 | 623 | 49 |
| FF100 | US | Jul/1971–May/2023 | 623 | 100 |
| FF100MEOP | US | Jul/1971–May/2023 | 623 | 100 |

"asset" corresponds to a portfolio formed by sorting U.S. stocks based on firm-specific characteristics. Specifically, FF25 and FF32 are constructed based on book-to-market ratio (BE/ME) and investment level; FF49 includes 49 industry-based portfolios; FF100 is formed using market equity (ME) and BE/ME; and FF100MEOP uses ME and operating profitability. We exclude FF25EU from our experiments because it contains only 390 monthly records—significantly fewer than the 622 records in the other datasets—making it difficult to accumulate sufficient search steps for robust factor evolution and performance evaluation.

### B.2  MARKET DATA

We construct multiple datasets to evaluate our framework across different market environments. Specifically, we build a Hong Kong set (Table 5) and a US50 dataset (Table 6). For broader index-based studies, we additionally collect constituents of the NASDAQ100, CSI300, and CSI500 indices as of May 2025. These datasets allow us to examine performance in both regional and large-scale global markets.

For the Hong Kong market, our asset universe is based on the Hang Seng Technology Index constituents as of April 2025. Since several companies were listed relatively late and lack sufficient

historical records, we replace them with comparable firms from the same industry within the broader HSI index. To further reduce potential bias arising from the underperformance of the Hong Kong market before mid-2024, we supplement the pool with 15 randomly selected large-cap blue-chip stocks from the HSI index. Importantly, this supplementation is carried out without using past performance as a selection criterion, ensuring that the resulting universe remains statistically balanced, representative of the market, and neutral for evaluation.

In some cases, company listings post-date the beginning of our data period, leading to insufficient coverage for backtesting. To maintain robustness, we exclude such firms from the datasets. For example, in constructing the CSI300 market set, we initially include all 300 constituent companies. However, 13 firms have IPO dates too recent to provide adequate historical data. These are removed, resulting in a final universe of 287 stocks. Given that the excluded firms account for only a small fraction of the index (13 out of 300), the overall impact on statistical validity and representativeness is minimal. This treatment of index constituents ensures consistency across datasets while preventing distortions caused by data incompleteness. By carefully handling late listings and supplementing underrepresented markets, we construct asset pools that are both practical for empirical analysis and faithful to real-world market structures.

Table 5: Hong Kong Stock Tickers with Full Company Names in the HSI Dataset

| Ticker | Company Name | Ticker | Company Name | Ticker | Company Name |
|--------|--------------|--------|--------------|--------|--------------|
| 1928.HK | SANDS CHINA LTD | 9868.HK | XPeng Inc. | 9633.HK | Nongfu Spring |
| 1929.HK | CHOW TAI FOOK | 0285.HK | BYD ELECTRONIC | 9618.HK | JD.com, Inc. |
| 2382.HK | Sunny Optical | 0388.HK | HKEX | 0939.HK | CCB |
| 1024.HK | KUAISHOU-W | 1810.HK | XIAOMI-W | 3690.HK | Meituan |
| 0241.HK | ALI HEALTH | 9626.HK | Bilibili Inc. | 0268.HK | KINGDEE INT'L |
| 3888.HK | Kingsoft Corp. | 0981.HK | SMIC | 9992.HK | Pop Mart |
| 9988.HK | Alibaba Group | 0005.HK | HSBC HOLDINGS | 1088.HK | CHINA SHENHUA |
| 1398.HK | ICBC | 9999.HK | NetEase, Inc. | 0941.HK | CHINA MOBILE |
| 0522.HK | ASMPT | 3908.HK | CICC | 0700.HK | TENCENT |
| 9961.HK | Trip.com | 1347.HK | HUA HONG SEMI | 1211.HK | BYD COMPANY |
| 0020.HK | SENSETIME | 2899.HK | Zijin Mining | 2338.HK | Weichai Power |
| 0992.HK | LENOVO GROUP | 2015.HK | Li Auto Inc. | 6690.HK | Haier Smart Home |
| 6066.HK | CSC Financial | 0772.HK | CHINA LIT | 6618.HK | JD Health |
| 9888.HK | Baidu | 2318.HK | Ping An Insurance | 0780.HK | TONGCHENG TRAVEL |
| 0945.HK | MANULIFE | 0386.HK | SINOPEC CORP | 3328.HK | Bank of Communications |

Table 6: U.S. Stock Symbols with Exact Company Names of US50 Dataset

| Ticker | Company Name | Ticker | Company Name | Ticker | Company Name |
|--------|--------------|--------|--------------|--------|--------------|
| AAPL | Apple | ADI | Analog Devices | ADBE | Adobe |
| ADP | Automatic Data Processing | ADSK | Autodesk | AMD | Advanced Micro Devices |
| AMAT | Applied Materials | AMGN | Amgen | AMZN | Amazon |
| APP | Applovin | ASML | ASML Holding | AVGO | Broadcom |
| BA | Boeing | BKNG | Booking Holdings | CDNS | Cadence Design Systems |
| CMCSA | Comcast | COST | Costco | CSCO | Cisco Systems |
| CSX | CSX Corporation | CTAS | Cintas Corporation | FTNT | Fortinet |
| GILD | Gilead Sciences | GOOG | Alphabet (Class C) | BRK-B | Berkshire Hathaway (Class B) |
| HON | Honeywell | INTC | Intel | INTU | Intuit |
| ISRG | Intuitive Surgical | KLAC | KLA Corporation | LIN | Linde plc |
| LRCX | Lam Research | MELI | MercadoLibre | META | Meta Platforms |
| MRVL | Marvell Technology | MSFT | Microsoft | MU | Micron Technology |
| NFLX | Netflix | NVDA | NVIDIA | PANW | Palo Alto Networks |
| PEP | PepsiCo | PLTR | Palantir | QCOM | Qualcomm |
| ROP | Roper Technologies | SBUX | Starbucks | SNPS | Synopsys |
| TMUS | T-Mobile US | TSLA | Tesla | TSM | TSMC |
| TXN | Texas Instruments | VRTX | Vertex Pharmaceuticals | | |

## C  PROMPTS DESIGN

To ensure reproducibility and consistency, we design a structured system prompt that explicitly constrains the LLM's action space and output format. The prompt specifies the role of the model, strict coding rules (e.g., Python-only, NumPy operations, standardized naming), and clear criteria for factor improvement through mutation or crossover. By enforcing these requirements, we guarantee that all generated factors are executable, comparable across versions, and aligned with the intended evaluation framework. As shown in Figure 7, the system prompt enforces strict coding and naming standards, guiding the LLM to generate reproducible and robust factor definitions.

---

**Prompt: System Prompts for Factor Generation**

```
You are a world-class quantitative researcher and Python programmer
specializing in alpha factor design for asset ranking.
Your task is to generate high-quality Python factor functions that
are evolved versions of provided factors.
STRICT REQUIREMENTS:
1.  Output ONLY a Python list of function strings - no comments or
explanations
2.  Each function MUST:
  - Be bug-free and executable
  - Maintain identical input signature:  prices, window
  - Use only numpy (as np), don't depend on any external function or
variable, you need to do computation all inside function
  - Handle edge cases (short series, NaNs)
  - Clearly indicate if combining or modifying existing factors
3.  Absolute prohibitions:
  - No external functions
  - No hardcoded values that should be parameters
  - No pandas or other libraries
  - No comments in output code
4.  Factor name rules:  [factor_name_part]_[window_size]_v[version
number], the window_size can only be the following value:  3, 7, 14,
21
5.  Value of output factor:  For factors, higher value means better
asset, please make sure the output value is positive related to
performance of assets.
ACTION SPACE:
1.  Improve existing factors by mutation:
  - Modifying parameters (e.g., inner parameters)
  - Adjusting logic
  - Updating inside operators for factors
2.  Improve existing factors by crossover:
  - Combining two existing factors to create a new one if you think
they can work together
  - Restart version number from v1 for new factors
IMPROVEMENT CRITERIA:
1.  Version increments must show clear:
  - if you improve from a given version, increase 1 to version
number, the version number can only be integer like v1, v2, v3,
don't include any other character.
  - Performance enhancement
  - Robustness improvement
  - Computational efficiency
  - if you create a new factor by crossover from other two,
restart version number from v1, and use the name like:
factor1_comb_factor2, where factor1 and factor2 are the names of
the two factors you combined.
2.  Combined factors should demonstrate:
  - Logical interaction
  - Complementary strengths
  - Better risk-adjusted returns
```

```
   - Don't make combined factors too complex, try to keep it simple
and easy to understand.
For string version of function, you should be very careful about
format of python, for example:
"def test_run_avg(*args, **kwargs):\n a=np.array(prices)\n if a:\n
print(a)\n return np.mean(kwargs['prices'])\n"
Be careful of symbol split the line, the different space of tab for
if statement.
Output example:
["def momentum_7_v3(prices, window=10):  return ...",
 "def breakout_comb_meanrevert_21_v1(prices, window=20):  return ..."]
```

Figure 7: System prompt used to guide LLMs in generating and evolving alpha factors. The design enforces strict coding standards, naming conventions, and improvement criteria to ensure reproducibility and reliability.

To guide large language models effectively, we design structured user prompts in Figure 8 that clearly convey task objectives, input formats, and expected outputs. Our prompts incorporate both contextual background and explicit instructions, ensuring the model can interpret factor expressions, evaluate their quality, and generate consistent results.

---

**Prompt: LLM-Generated Factor Optimization**

```
You are a professional quantitative researcher.
Design and optimize alpha factors using ONLY price data.

Existing Library Factors:
<str_lib_factor>

Previous LLM-Generated Factors:
<str_gen_factor>

Recent Performance Metrics:
<recent_performance>
```

Figure 8: Structure of the user prompt where the model receives existing library factors, previously generated factors, and recent performance metrics as input.

---

**Sample of Performance and Factor Quality Information**

**Performance Information:**

| Factor | mean_return | std_return | sharpe_ratio | max_drawdown | final_value |
|---|---|---|---|---|---|
| max_min_ratio_comb_momentum_3_v2 | 0.00375 | 0.01335 | 0.28047 | -0.05990 | 124.48487 |
| drawdown_comb_sharpe_14_v5 | 0.00360 | 0.01242 | 0.28959 | -0.03885 | 123.48065 |
| drawdown_comb_sharpe_14_v2 | 0.00360 | 0.01242 | 0.28959 | -0.03885 | 123.48065 |
| drawdown_comb_sharpe_14_v1 | 0.00360 | 0.01242 | 0.28959 | -0.03885 | 123.48065 |
| drawdown_comb_sharpe_14_v3 | 0.00360 | 0.01242 | 0.28959 | -0.03885 | 123.48065 |

**Factor Quality Information:**

| Factor | mean_rankic | std_rankic | mean_recall@20 | std_recall@20 |
|---|---|---|---|---|
| drawdown_comb_momentum_14_v6 | 0.055634 | 0.215905 | 0.417500 | 0.107170 |
| sharpe_ratio_14 | 0.047617 | 0.192138 | 0.405833 | 0.123522 |
| drawdown_comb_momentum_comb_rsi_14_v1 | 0.047588 | 0.214157 | 0.421667 | 0.103427 |
| drawdown_comb_momentum_comb_rsi_14_v2 | 0.047588 | 0.214157 | 0.421667 | 0.103427 |
| sharpe_ratio_14_v5 | 0.047463 | 0.191842 | 0.405833 | 0.123522 |

Figure 9: Sample data in prompts of factor evaluation, illustrating both performance information (returns, risk, and drawdown metrics) and factor quality information (RankIC stability and recall measures). This provides the LLM with concrete signals for guiding further factor optimization.

## D    EXPERIMENT SETTINGS

### D.1    HYPERPARAMETERS AND SETTINGS

For our experiment, for all methods, we don't include transaction cost; we use default hyper parameters for mSSRM-PGA Lin et al. (2024a)[1] and ASMCVaR Lin et al. (2024b)[2] from their open-source code.

For machine learning-based sparse portfolio strategies using XGBoost and LightGBM, at each decision step, we train a model on sliding windows of historical price and return features, with periodic retraining every fixed number of steps to simulate realistic model updating. The model outputs asset-level scores, which are normalized and used to select the top-$m$ assets with equal weighting.

Due to the nature of our algorithm, for all experiments, we compute the daily portfolio return using a 1/N baseline strategy during the warm-up phase—i.e., from the start point until the first evolution step begins. In contrast, for optimization-based baselines, since these methods can generate results based on a small historical window, we evaluate them from the very first step, assuming historical data are already available.

We choose two common online LLM services: GPT-4.1 and DeepSeek-V3. In our experiments, we adopt a fixed lookback window of $T = 30$ for computing alpha factors, as prior work (Lin et al., 2024a) and our tests show minimal sensitivity to window size. Moreover, most factor designs (e.g., momentum, mean reversion) rely on short-term price patterns, making longer windows unnecessary. Considering implementation constraints and LLM context limits, we use only two variables—closing price and returns—for factor construction. All factor scores are normalized to the range [-1, 1], and we select top-5 scoring assets at each step. To isolate factor quality, we apply equal weighting to selected assets throughout.

### D.2    AGGREGATED EVALUATION IN MARKET DATASET

Given the use of online LLM services and the volatile nature of financial markets, the overall performance of our framework may vary across runs due to three main factors: (1) stochastic quality of LLM outputs, (2) occasional market events that align with certain factors, and (3) the compounding effects across long backtesting periods. Specifically, for each market dataset, to mitigate noise from individual LLM outputs and transient market conditions, we repeat the factor search process three times and report performance under *aggregated evaluation*, where all discovered factors across the three runs are pooled together into a unified factor library, and then re-evaluated in a single backtest.

### D.3    SETTINGS FOR EVALUATION METRIC

We assume the initial portfolio value in all experiments is $P_0 = 1$, so cumulative wealth (CW) is equivalent to the final value, i.e., $\text{CW} = P_T/P_0 = P_T$. When computing the Sharpe Ratio, we set the risk-free rate to zero. This choice is motivated by three factors: (1) key baselines, such as mSSRM-PGA and ASMCVaR, also assume a zero rate, ensuring alignment; and (2) our methods consistently achieve high CW and Sharpe Ratios, making the effect of a nonzero rate negligible in practice.

### D.4    DATA SAFETY GUARANTEE

To ensure the integrity and fairness of our experimental setup, we strictly control the information provided to the LLM. Specifically, we only expose the model to abstract *factor definitions* and their corresponding *backtest performance metrics* (e.g., rank IC, Sharpe ratio), without revealing any specific stock identifiers, price series, or temporal indicators (e.g., dates or market phases). During the evaluation phase, we execute the portfolio backtest based solely on the factors generated in each search iteration, without involving the LLM in any subsequent computation or refinement. This design eliminates the possibility of data leakage and ensures that the backtest performance is unaffected by any external knowledge, including information potentially known to the LLM before its training cutoff date. Our pipeline thus guarantees a strict separation between the data used for search and the unseen future used for evaluation.

---

[1]https://github.com/laizhr/mSSRM-PGA
[2]https://github.com/linyizun2024/ASMCVaR

# E    ADDITIONAL DETAILS IN EXPERIMENTS

## E.1    DISCUSSION ON BENCHMARK DATASET

Table 7 provides a clear comparison across the five Fama–French datasets. Within the baselines, the $1/N$ portfolio delivers stable but limited results, reflecting its role as a market proxy. SSPO performs poorly across all settings, showing that simple sparse optimization cannot adapt well to different market structures. Min-CVaR achieves higher Sharpe Ratios than $1/N$, but its cumulative wealth is relatively low, while Max-Sharpe consistently dominates traditional methods, achieving the best CW in most datasets. Still, its SR values only slightly exceed those of $1/N$, suggesting limited risk-adjusted improvement. For sparse machine learning models, LGBM and XGBoost provide moderate gains but are not competitive with optimization-based strategies. mSSRM-PGA shows stronger scalability and achieves reasonable balance between CW and SR, while ASMCVaR generally delivers the highest Sharpe Ratios among the baselines, though at the cost of weaker cumulative returns in larger universes.

In contrast, the proposed framework consistently achieves superior results. Under $m = 10$, it records the highest cumulative wealth across nearly all datasets: GPT-4.1 excels in FF25, FF100, and FF100MEOP, while DeepSeek achieves top performance in FF32 and FF49. Sharpe Ratios remain on par with or slightly above the best optimization-based competitors, confirming that the gains are not achieved by sacrificing stability. The advantage becomes especially pronounced in large universes such as FF100 and FF100MEOP, where CW exceeds 1800, more than double the values achieved by Max-Sharpe and ASMCVaR.

At larger cardinalities ($m = 15, 20$), the framework remains competitive, with score-to-weight allocation improving capital distribution while keeping sparsity intact. GPT-4.1 is particularly strong in larger universes, while DeepSeek provides more consistent SR values across datasets. Overall, the results highlight two main advantages: stronger cumulative wealth growth in both small and large universes, and robustness in risk-adjusted performance, showing that the proposed method scales better and adapts more effectively than traditional or machine learning baselines.

Table 7: Cumulative Wealth (CW↑) and Sharpe Ratio (SR↑) across five Fama-French benchmark datasets (FF25, FF32, FF49, FF100, FF100MEOP). Arrows indicate preferred direction of performance.

| Group | Method | FF25 | | FF32 | | FF49 | | FF100 | | FF100MEOP | |
|---|---|---|---|---|---|---|---|---|---|---|---|
| | | CW↑ | SR↑ | CW↑ | SR↑ | CW↑ | SR↑ | CW↑ | SR↑ | CW↑ | SR↑ |
| Baseline | 1/N | 374.13 | 0.229 | 452.84 | 0.225 | 254.67 | 0.207 | 389.44 | 0.210 | 371.33 | 0.209 |
| | SSPO | 76.70 | 0.140 | 15.37 | 0.100 | 62.27 | 0.114 | 1.21 | 0.046 | 9.97 | 0.086 |
| | Min-cVaR | 386.71 | 0.239 | 289.93 | 0.231 | 207.53 | 0.248 | 155.70 | 0.203 | 258.01 | 0.220 |
| | Max-Sharpe | **553.10** | **0.243** | **718.62** | **0.251** | 210.94 | 0.234 | **386.86** | **0.223** | **379.62** | **0.229** |
| m=10 | LGBM | 281.49 | 0.217 | 530.49 | 0.226 | 169.18 | 0.189 | 216.81 | 0.189 | 284.60 | 0.198 |
| | XGB | 292.22 | 0.220 | 551.77 | 0.225 | 354.46 | 0.206 | 392.18 | 0.200 | 269.10 | 0.193 |
| | mSSRM-PGA | 606.84 | 0.241 | 748.59 | 0.249 | 168.99 | 0.222 | 379.39 | 0.216 | 305.86 | 0.222 |
| | ASMCVaR | 638.19 | **0.252** | 670.40 | 0.244 | 409.18 | 0.224 | 491.12 | 0.228 | 405.38 | 0.220 |
| | DeepSeek | 639.66 | 0.251 | **923.52** | **0.244** | **692.91** | **0.228** | 1232.86 | 0.238 | 342.66 | 0.201 |
| | GPT4.1 | **708.34** | 0.255 | 613.92 | 0.232 | 565.07 | 0.222 | **1836.34** | **0.253** | **555.49** | **0.215** |
| m=15 | LGBM | 312.50 | 0.221 | 541.42 | 0.229 | 194.64 | 0.196 | 223.94 | 0.191 | 301.82 | 0.202 |
| | XGB | 308.41 | 0.223 | 491.32 | 0.223 | 377.05 | 0.214 | 364.16 | 0.201 | 355.87 | 0.204 |
| | mSSRM-PGA | 601.25 | 0.240 | 744.91 | 0.249 | 171.95 | 0.223 | 415.54 | 0.222 | 306.34 | 0.223 |
| | ASMCVaR | **676.35** | **0.252** | 690.99 | 0.243 | 526.84 | 0.235 | 523.49 | 0.230 | 518.80 | 0.227 |
| | DeepSeek | 546.03 | 0.246 | 715.55 | 0.239 | 586.14 | **0.231** | 983.84 | 0.233 | 371.13 | 0.206 |
| | GPT4.1 | 530.26 | 0.246 | 608.88 | 0.234 | **533.61** | 0.228 | **1289.85** | **0.244** | **565.41** | **0.217** |
| m=20 | LGBM | 336.90 | 0.225 | 468.51 | 0.225 | 207.05 | 0.200 | 249.65 | 0.194 | 332.51 | 0.205 |
| | XGB | 346.15 | 0.227 | 482.98 | 0.225 | 307.56 | 0.211 | 349.19 | 0.201 | 357.33 | 0.205 |
| | mSSRM-PGA | 601.26 | 0.240 | 744.91 | 0.249 | 171.97 | 0.223 | 422.39 | 0.223 | 304.77 | 0.223 |
| | ASMCVaR | **653.38** | **0.249** | 731.32 | 0.244 | 530.76 | 0.234 | 583.51 | 0.232 | 508.86 | 0.225 |
| | DeepSeek | 460.94 | 0.239 | 634.17 | 0.236 | **592.02** | **0.236** | 882.04 | 0.230 | 406.98 | 0.210 |
| | GPT4.1 | 458.54 | 0.239 | 580.23 | 0.234 | 509.44 | 0.231 | **1082.46** | **0.239** | **544.27** | **0.218** |

## E.2 ADDITIONAL RESULTS IN HONG KONG MARKET

The results on the HSI45 dataset highlight the advantage of our framework over both traditional baselines and state-of-the-art sparse portfolio models. For $m = 10$, **DeepSeek** achieves the highest cumulative wealth (CW = 3.463) and Sharpe ratio (SR = 0.080), clearly outperforming ASMCVaR (CW = 2.481, SR = 0.052) and all other methods. **GPT** also delivers strong returns (CW = 2.789, SR = 0.067), showing consistent gains compared to machine learning selectors such as LGBM and XGBoost. When the portfolio size increases to $m = 15$, our method continues to deliver competitive performance, with ASMCVaR achieving slightly higher CW (2.647), while our framework variants maintain higher Sharpe ratios and stable returns. Overall, the findings confirm that it provides more robust performance on the Hong Kong market, especially under stricter sparsity constraints.

Table 8: Evaluation of Cumulative Wealth (CW↑) and Sharpe Ratio (SR↑) on HSI45 dataset.

| Group | Method | CW↑ | SR↑ |
|---|---|---|---|
| | 1/N | 1.333 | 0.029 |
| Baseline | Min-cVaR | 1.628 | 0.063 |
| | Max-Sharpe | 1.428 | 0.043 |
| | LGBM | 1.611 | 0.038 |
| | XGBoost | 1.581 | 0.035 |
| | mSSRM-PGA | 0.766 | -0.003 |
| m=10 | ASMCVaR | 2.481 | 0.052 |
| | **DeepSeek** | **3.463** | **0.080** |
| | **GPT** | 2.789 | 0.067 |
| | LGBM | 1.588 | 0.037 |
| | XGBoost | 1.586 | 0.036 |
| | mSSRM-PGA | 0.766 | -0.003 |
| m=15 | ASMCVaR | **2.647** | 0.054 |
| | **DeepSeek** | 2.364 | 0.061 |
| | **GPT** | 2.277 | 0.058 |

## E.3 EXPERIMENT ON PORTFOLIO WEIGHT ALLOCATION

Beyond equal-weighted allocation, we further investigate a score-to-weight transformation mentioned in Table **??**, where factor scores are directly mapped into portfolio weights. Table 9 reports the results under this scheme across the five Fama-French benchmark datasets. We observe that applying score-to-weight allocation consistently enhances performance relative to equal weighting for both DeepSeek and GPT-4.1 variants. In particular, the cumulative wealth (CW) gains are substantial, with improvements most pronounced on larger universes such as FF100 and FF100MEOP. Sharpe Ratios (SR) also improve modestly or remain stable, suggesting that the gains are not achieved at the expense of increased volatility. These results highlight that incorporating factor intensity information into weight assignment provides an additional performance edge, complementing the benefits of sparse factor selection.

Table 9: Cumulative Wealth (CW↑) and Sharpe Ratio (SR↑) of our methods across five Fama-French benchmark datasets. Gray background highlights the extracted results.

| Group | Method | FF25 CW↑ | FF25 SR↑ | FF32 CW↑ | FF32 SR↑ | FF49 CW↑ | FF49 SR↑ | FF100 CW↑ | FF100 SR↑ | FF100MEOP CW↑ | FF100MEOP SR↑ |
|---|---|---|---|---|---|---|---|---|---|---|---|
| m=10 | DeepSeek | 639.66 | 0.251 | **923.52** | 0.244 | **692.91** | 0.228 | 1232.86 | 0.238 | 342.66 | 0.201 |
| | + Scores to Weights | 1064.56 | 0.254 | 797.71 | 0.233 | 441.48 | 0.187 | 1464.97 | 0.240 | 359.86 | 0.201 |
| | GPT 4.1 | 708.34 | 0.255 | 613.92 | 0.232 | 565.07 | 0.222 | 1836.34 | 0.253 | **555.49** | 0.215 |
| | + Scores to Weights | **1408.44** | 0.273 | 847.35 | 0.234 | 593.18 | 0.200 | **2434.51** | 0.257 | 461.87 | 0.203 |
| m=15 | DeepSeek | 546.03 | 0.246 | 715.55 | 0.239 | 586.14 | 0.231 | 983.84 | 0.233 | 371.13 | 0.206 |
| | + Scores to Weights | 1049.53 | 0.254 | 735.64 | 0.231 | 427.20 | 0.187 | 1353.38 | 0.239 | 343.86 | 0.200 |
| | GPT 4.1 | 530.26 | 0.246 | 608.88 | 0.234 | 533.61 | 0.228 | 1289.85 | 0.244 | **565.41** | 0.217 |
| | + Scores to Weights | **1364.54** | 0.272 | **840.14** | 0.234 | **638.79** | 0.203 | **2060.59** | 0.254 | 455.94 | 0.204 |
| m=20 | DeepSeek | 460.94 | 0.239 | 634.17 | 0.236 | 592.02 | 0.236 | 882.04 | 0.230 | 406.98 | 0.210 |
| | + Scores to Weights | 1029.55 | 0.253 | 721.73 | 0.231 | 412.48 | 0.187 | 1320.88 | 0.240 | 338.07 | 0.200 |
| | GPT 4.1 | 458.54 | 0.239 | 580.23 | 0.234 | 509.44 | 0.231 | 1082.46 | 0.239 | **544.27** | 0.218 |
| | + Scores to Weights | **1334.50** | 0.271 | **847.87** | 0.234 | **642.53** | 0.204 | **1980.99** | 0.253 | 446.04 | 0.204 |

To further validate our factor-based approach, we extend beyond equal-weight allocation and convert raw factor scores into portfolio weights under a sparsity constraint of $m = 10$ assets. Figure 18b illustrates the temporal evolution of portfolio values against three benchmarks: (1) the passive market baseline, (2) the optimization-based ASMCVaR model, and (3) our factor search regimes with score-to-weight transformation. The results show that our weighting scheme consistently outperforms both benchmarks across market cycles, preserving capital during the 2022 downturn while

participating strongly in subsequent bull markets. This effectiveness arises from the dynamic concentration mechanism, which diversifies exposure under high volatility but shifts aggressively toward top-ranked assets during rallies.

We also examine the robustness of this weighting scheme by varying the temperature parameter $\tau$ in the score-to-weight mapping. As reported in Table 10, lower $\tau$ values (0.1–1.0) yield stable improvements across both US50 and HSI45 datasets, maintaining balanced Sharpe Ratios with moderate cumulative wealth. However, at higher settings (e.g., US50, $m = 10$, $\tau = 2.0$), the portfolios become extremely concentrated, producing explosive terminal wealth (e.g., CW=389.01 for DeepSeek) but at the cost of substantial tail risk. This outcome reflects the model's tendency to allocate nearly all capital to a few assets that happened to experience large upward moves, amplifying gains through compounding but sacrificing robustness. These findings indicate that while aggressive weighting can exploit rare opportunities, it also introduces fragility. In practice, equal-weighted or temperature-smoothed schemes ($\tau \in [0.5, 1.0]$) offer a more reliable balance, achieving strong performance while mitigating exposure to extreme concentration and overfitting.

Table 10: Portfolio performance (Cumulative Wealth (CW) / Sharpe Ratio (SR)) using score-to-weight allocation under different temperature settings.

| $m$ | Model | US50 | | | | HSI45 | | | |
| --- | --- | --- | --- | --- | --- | --- | --- | --- | --- |
| | | temp=0.1 | temp=0.5 | temp=1.0 | temp=2.0 | temp=0.1 | temp=0.5 | temp=1.0 | temp=2.0 |
| 10 | DeepSeek | 179.554 / 0.142 | 247.376 / 0.147 | 318.868 / 0.149 | 389.010 / 0.148 | 6.104 / 0.096 | 6.246 / 0.095 | 5.876 / 0.089 | 4.901 / 0.078 |
| | GPT-4.1 | 132.178 / 0.151 | 180.417 / 0.155 | 245.845 / 0.157 | 332.340 / 0.154 | 5.758 / 0.087 | 6.084 / 0.087 | 6.545 / 0.087 | 7.486 / 0.088 |
| 15 | DeepSeek | 153.509 / 0.137 | 220.427 / 0.144 | 293.589 / 0.147 | 371.420 / 0.147 | 5.494 / 0.092 | 5.884 / 0.093 | 5.682 / 0.088 | 4.837 / 0.078 |
| | GPT-4.1 | 108.623 / 0.146 | 150.682 / 0.150 | 211.567 / 0.153 | 303.639 / 0.152 | 4.458 / 0.077 | 4.984 / 0.080 | 5.613 / 0.082 | 6.792 / 0.085 |

### E.4 PORTFOLIO PERFORMANCE UNDER TRANSACTION COST

We conduct experiments by applying transaction costs on portfolio changes to assess the real-world robustness of our strategy. Specifically, we evaluate two widely-used cost settings: $c = 0.1\%$ and $c = 0.2\%$, applied to each reallocation step. The experiments are conducted under $m = 10$ across US50, HSI45, and CSI300 datasets.

As shown in Table 11, incorporating transaction costs significantly affects cumulative wealth (CW), particularly under higher cost levels. Nevertheless, our method maintains strong performance, especially on US50 and CSI300, where the net returns remain competitive even after cost deductions. This demonstrates that the generated factors are not overly sensitive to short-term noise and still capture meaningful market signals.

We attribute the performance drop primarily to the use of an equal-weighting strategy, which does not optimize for turnover and may result in frequent portfolio shifts. Without additional sparsity constraints on asset selection, the top-$k$ assets can change substantially at each step, amplifying transaction costs. Future improvements could include turnover-aware selection mechanisms. Since factor scores for top-ranked assets are often close in value, a holding-aware filtering step could be applied to prioritize assets already in the portfolio, thus reducing unnecessary trades while preserving performance.

Table 11: Backtest results under transaction costs $c = 0.1\%$ and $c = 0.2\%$. Metrics shown are Cumulative Wealth (CW), Sharpe Ratio (SR), and Maximum Drawdown (MDD) across datasets US50, HSI45, and CSI300.

| $c$ | Method | US50 | | | HSI45 | | | CSI300 | | |
| --- | --- | --- | --- | --- | --- | --- | --- | --- | --- | --- |
| | | CW | SR | MDD | CW | SR | MDD | CW | SR | MDD |
| - | GPT 4.1 | 39.746±15.484 | 0.154±0.013 | 0.252±0.021 | 3.338±0.770 | 0.076±0.012 | 0.324±0.041 | 3.862±0.802 | 0.086±0.011 | 0.290±0.079 |
| | DeepSeek | 32.709±6.244 | 0.149±0.003 | 0.261±0.014 | 3.203±0.906 | 0.076±0.015 | 0.385±0.024 | 2.451±0.412 | 0.060±0.010 | 0.356±0.022 |
| 0.1% | GPT 4.1 | 25.293±10.802 | 0.135±0.016 | 0.256±0.021 | 2.710±0.571 | 0.065±0.011 | 0.340±0.049 | 2.624±0.425 | 0.064±0.008 | 0.354±0.093 |
| | DeepSeek | 22.058±4.053 | 0.133±0.003 | 0.265±0.015 | 2.643±0.775 | 0.065±0.015 | 0.410±0.023 | 1.668±0.227 | 0.039±0.008 | 0.434±0.012 |
| 0.2% | GPT 4.1 | 16.114±7.464 | 0.117±0.018 | 0.260±0.022 | 2.201±0.422 | 0.054±0.011 | 0.358±0.056 | 1.785±0.208 | 0.043±0.006 | 0.412±0.103 |
| | DeepSeek | 14.876±2.646 | 0.117±0.003 | 0.270±0.016 | 2.182±0.665 | 0.054±0.016 | 0.433±0.022 | 1.136±0.130 | 0.018±0.007 | 0.502±0.009 |

### E.5 INFLUENCE OF FACTOR NUMBER

Figure 10 illustrates portfolio performance on the US50 and HSI45 datasets as the number of top factors included in construction is varied. This analysis examines how many factors should be retained to balance return and stability. Cumulative wealth (CW) remains consistently high when

using up to 10 factors, highlighting the strong ranking ability of the factor evolution process. The 15-asset portfolio (orange line) is less sensitive to the number of factors than the 10-asset case, likely due to diversification mitigating the effect of weaker signals. Across most configurations, even the lowest results remain above traditional baselines (dashed ASMCVaR lines), with the only exception being HSI45 under $m = 15$, where performance occasionally converges to the baseline. Meanwhile, RankIC curves remain stable across factor counts, indicating consistent predictive power of the selected factors.

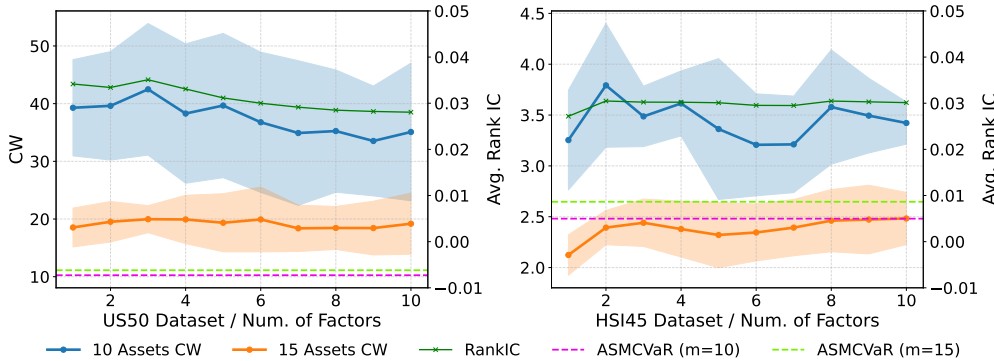

Figure 10: Cumulative Wealth (CW) and RankIC metrics on the US50 dataset using GPT-4.1. Results are reported across factor counts (1–10) and asset sizes. Solid lines denote mean CW, shaded areas indicate one standard deviation, and dashed lines represent ASMCVaR baselines.

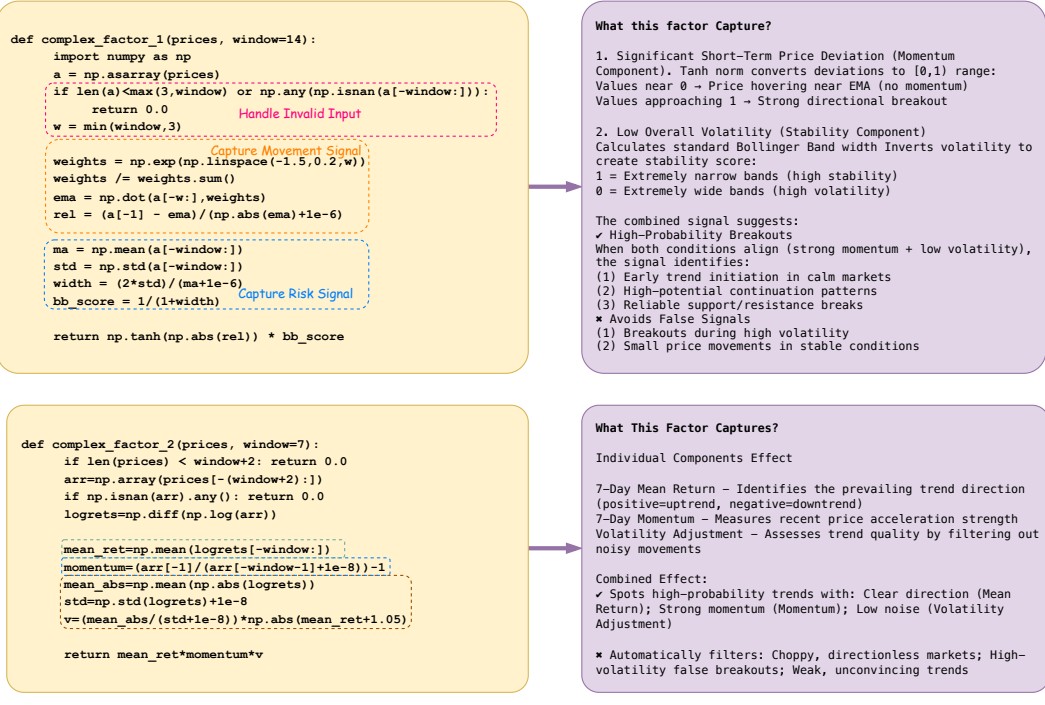

Figure 11: Examples of complex alpha factors generated by LLMs with their components and interpretability analysis. (Top) Combines EMA-normalized momentum with inverted Bollinger Band width to detect breakouts in stable markets. (Bottom) Fuses 7-day mean return, normalized momentum, and volatility-adjusted weighting to capture trend strength while reducing noise. Right-hand panels summarize factor logic and sub-component contributions, illustrating how LLMs compose expressive, interpretable signals.

# F FURTHER ANALYSIS OF BEHAVIORS

## F.1 ANALYSIS OF BEHAVIORS IN FACTOR GENERATION

Compared to traditional symbolic regression or heuristic search methods, large language models (LLMs) offer a distinct advantage in enabling end-to-end generation of executable factor code. This paradigm shift allows LLMs to fully leverage their generative capabilities by producing directly usable and structurally diverse alpha factors in a single step, without requiring predefined templates or restricted operator sets. Moreover, we observe that LLMs demonstrate remarkable creativity in generating complex, composite factors that are difficult to discover using conventional methods. These include multi-component expressions that fuse statistical indicators, momentum signals, risk adjustments, and stability metrics into a single function.

Throughout the evolutionary process, we identified several notable behavioral patterns of LLMs. First, LLMs frequently perform fine-grained *hyperparameter tuning* within existing factor structures, subtly adjusting exponents or scaling constants to optimize behavior. Second, they exhibit the ability to *fuse distinct factor structures*, effectively performing crossover operations by combining the logic of unrelated base signals into new composite forms.

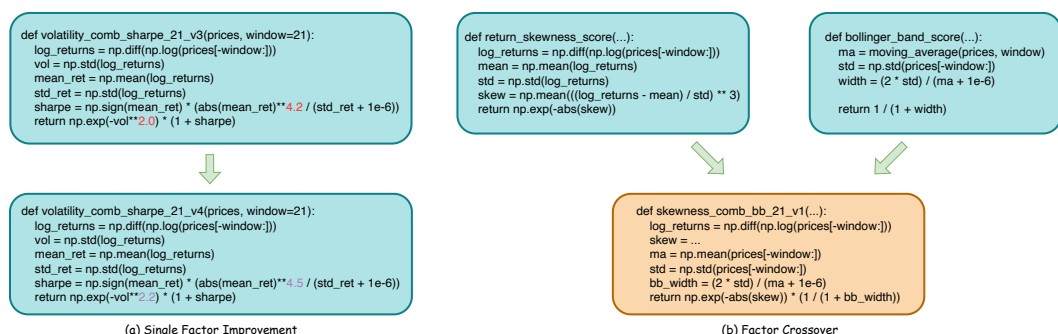

(a) Single Factor Improvement

(b) Factor Crossover

Figure 12: Illustration of different behaviors in LLM-guided factor generation. (a) shows a single-factor improvement where the LLM adjusts internal hyperparameters (e.g., exponent coefficients) to optimize factor behavior while preserving its structural form. (b) demonstrates a factor crossover operation, where two structurally distinct base factors (skewness and Bollinger band) are combined into a new composite expression. These examples highlight the LLM's capacity to explore both local refinements and global recombinations within the factor expression space.

In addition to structural flexibility, the generated factors often exhibit strong interpretability. As shown in Figure 11, many factors encode intuitive trading logic, such as momentum confirmation with volatility filtering or breakout detection under low-noise conditions. The factor pool also covers a wide range of styles and statistical properties, indicating that LLMs can adaptively utilize different types of signals.

By analyzing factor compositions across different time periods, we observe that the LLMs are able to produce *environment-aware* factors. For instance, during bull markets, factors emphasize trend continuation; in bear markets, they shift toward mean reversion or downside risk control; and in sideways markets, they focus on noise filtering or breakout identification. This adaptability suggests that LLMs inherently capture temporal regime features, and tailor factor expressions accordingly.

## F.2 PORTFOLIO ANALYSIS

In this section, we analyze how the profits were generated in our framework. We first examine the cumulative return curves across three datasets, as shown in Figure 16. Our LLM-generated portfolios exhibit two critical advantages: (1) the ability to preserve value during bear markets, maintaining stability even under broad market drawdowns, and (2) the capacity to amplify returns during bull markets by timely capturing growth opportunities. These traits demonstrate strong adaptability and robustness in diverse market regimes.

To further investigate how these profits were achieved, we analyze the asset composition of the portfolios. Figure 14 displays the most frequently selected tickers over the entire backtesting period, while Figure 15 shows detailed selection breakdowns for specific years in the US50 and HSI45

datasets. We observe that the investment strategy consistently identifies high-potential assets. For instance, in the US market, despite the significant downturn in 2022, our model frequently selected stocks such as GILD (Gilead Sciences), which exhibited strong counter-trend performance, and BRK-B (Berkshire Hathaway), known for its defensive stability. This shows that even in adverse macroeconomic conditions, our factor discovery mechanism is able to uncover resilient or contrarian opportunities.

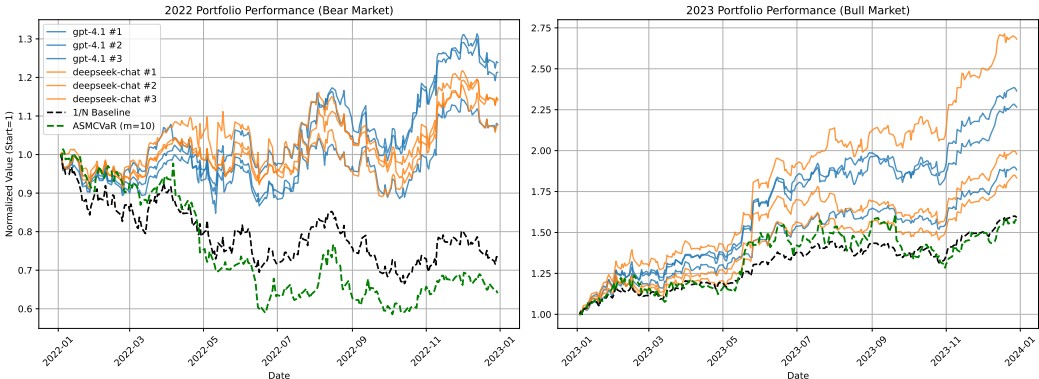

(a) US50 Performance in Bull (2023) and Bear Markets (2022)

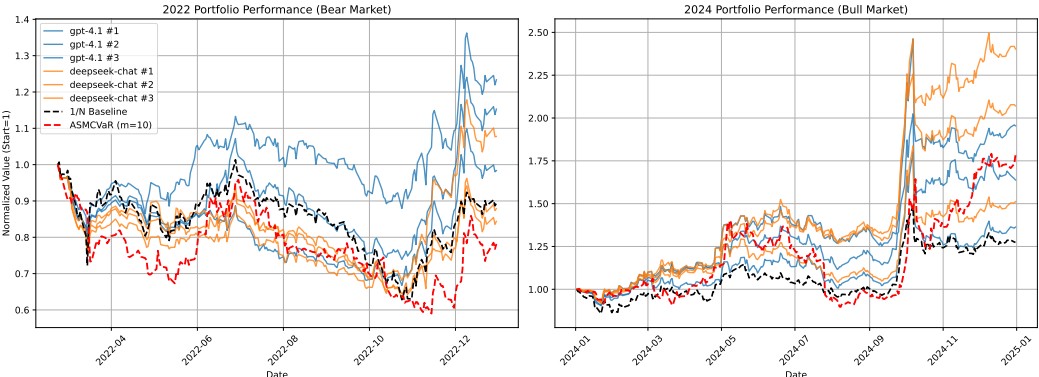

(b) HSI45 Performance in Bull (2024) and Bear Markets (2022)

Figure 13: Individual portfolio performance comparison during bull and bear market phases. These plots highlight the performance sensitivity of different strategies to market regimes.

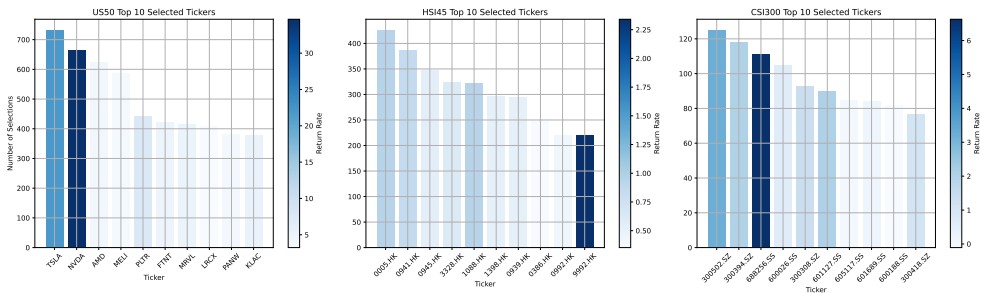

Figure 14: Top 10 most frequently selected assets by our method over the full backtest period for each market: US50 (left), HSI45 (middle), and CSI300 (right). Bar height indicates the number of selection occurrences, while the color shade reflects the cumulative return of each asset during the period. This highlights the model's preference for consistently high-performing stocks.

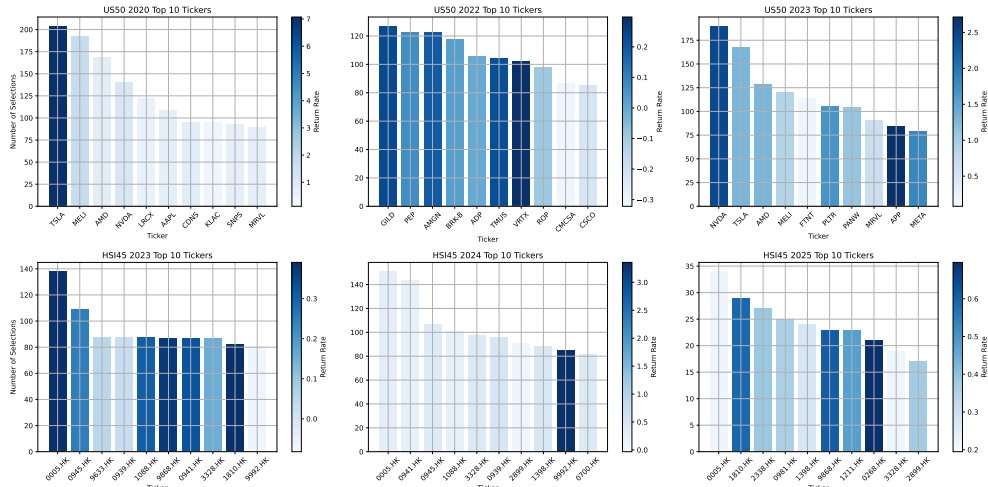

Figure 15: Annual snapshots of the top 10 most frequently selected assets under our framework in representative years for US50 (top row: 2020, 2022, 2023) and HSI45 (bottom row: 2023, 2024, 2025). Bar height denotes the number of selections within the given year, while the color encodes the annual return rate of each asset. This illustrates our method's dynamic asset preference and adaptability to varying market environments.

Moreover, in 2023, without any explicit macroeconomic input, the model captured the semiconductor boom by selecting NVDA and AMD, both are key players in the AI and hardware surge—demonstrating implicit awareness of sectoral trends. In Hong Kong markets, similar patterns are observed. During 2022–2023, when the HSI experienced a prolonged downturn, the model favored conservative financial assets such as 0005.HK (HSBC). As the market rebounded in late 2023 and 2024, the portfolios began shifting toward growth-oriented stocks like 9992.HK (Pop Mart), aligning with sentiment-driven and thematic plays in consumer markets.

These results reflect that our LLM-guided factor evolution exhibits behavior analogous to that of sophisticated human investors—favoring defense in downturns and seeking growth in recovery phases—without being explicitly trained on macroeconomic indicators. The emerging portfolio patterns reveal that our method not only adapts to market regimes but also autonomously captures meaningful economic signals through factor selection.

### F.3 FACTOR SCORES ANALYSIS

To better understand the underlying dynamics of factor behaviors throughout the investment horizon, we visualize the temporal evolution of factor scores in Figures 17a, 17b, and 17c, covering the US50, HSI45, and CSI300 datasets respectively. From these heatmaps, several key insights emerge:

- **i. Dynamic Adaptation to Market Regimes.** The distribution and sparsity patterns of factor scores vary significantly across different market stages, indicating that the LLM-generated factors are not static. For instance, in bearish regimes (e.g., early 2022 for CSI300), most factor scores are clustered near neutral or low values, whereas during bullish periods (e.g., late 2023 for HSI45), stronger signals emerge.

- **ii. Consistency of High-Scoring Factors.** We observe temporal continuity in certain high-scoring factors, often aligned with stocks that exhibit sustained uptrends. This reflects the model's ability to retain momentum information across time without explicit memory mechanisms.

- **iii. Emergent Sparsity Patterns.** Across datasets, factor sparsity appears to adjust naturally. For example, during volatile or uncertain market conditions, fewer dominant factors appear, suggesting a cautious allocation strategy. Conversely, in more stable or trending markets, factor scores become denser and more directional.

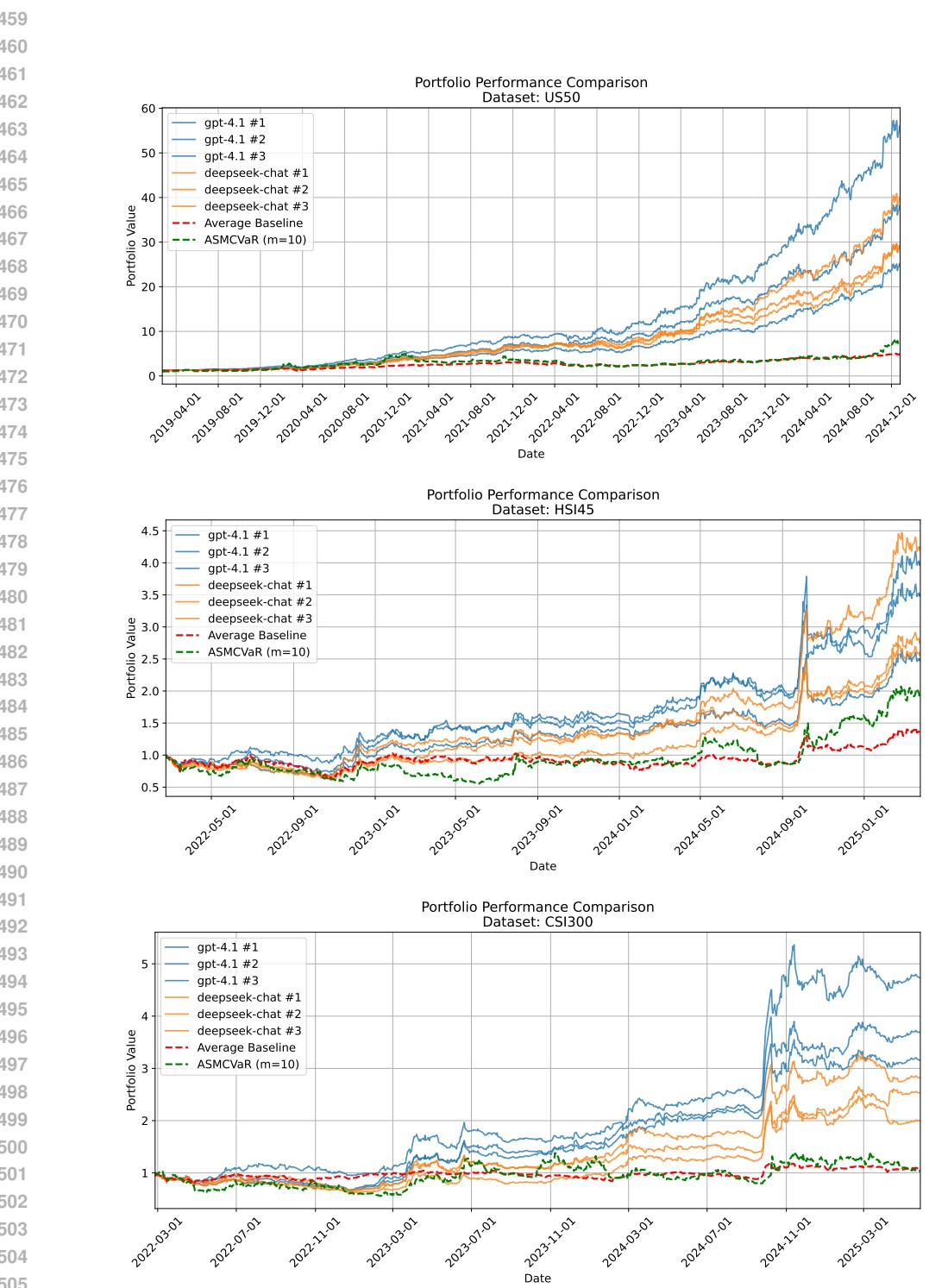

Figure 16: Portfolio performance comparison across US50, HSI45, and CSI300 datasets. Each plot shows the evolution of LLM-generated portfolios versus baselines and the ASMCVaR benchmark over time.

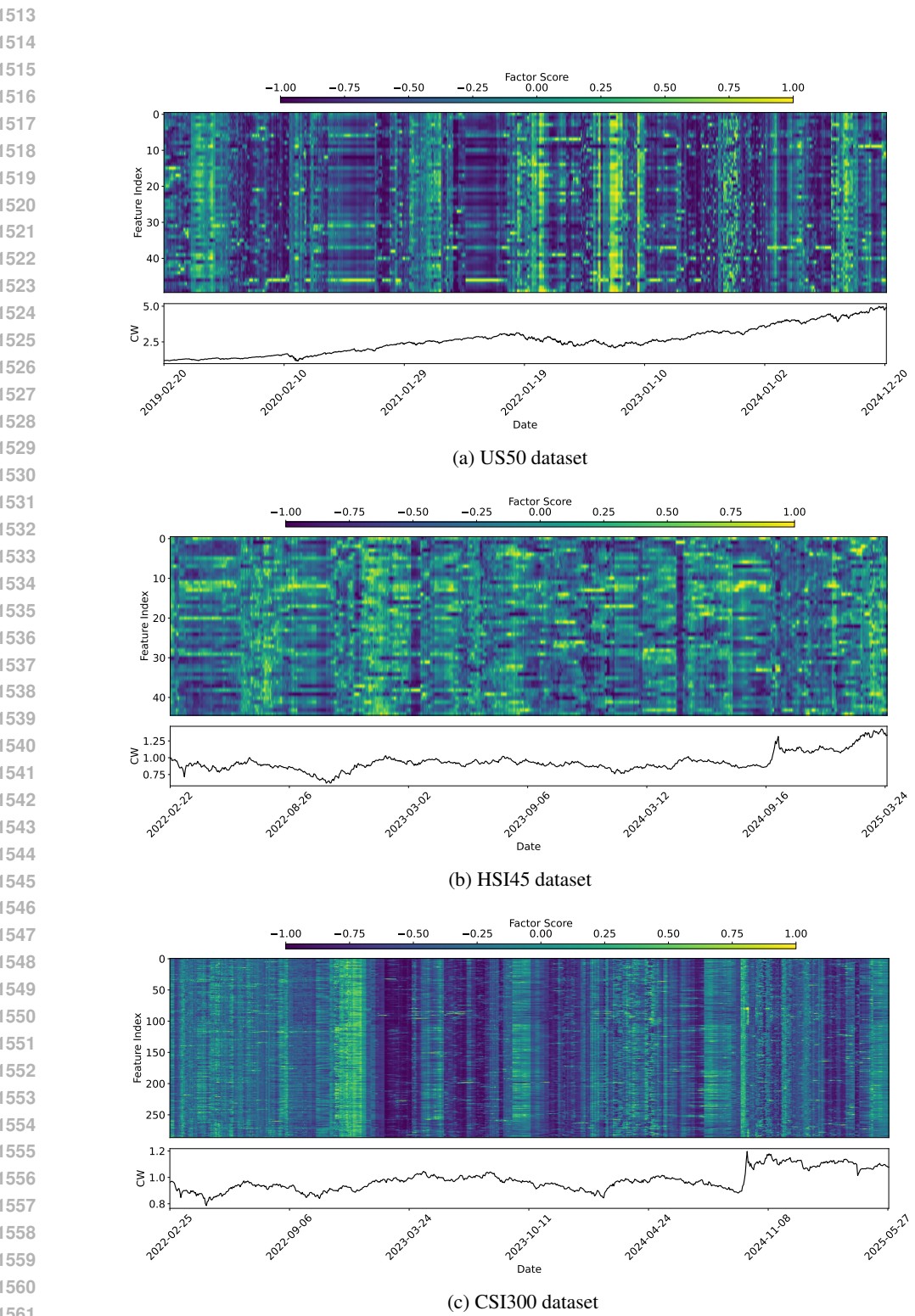

(a) US50 dataset

(b) HSI45 dataset

(c) CSI300 dataset

Figure 17: Factor score heatmaps and corresponding baseline curves across three datasets.

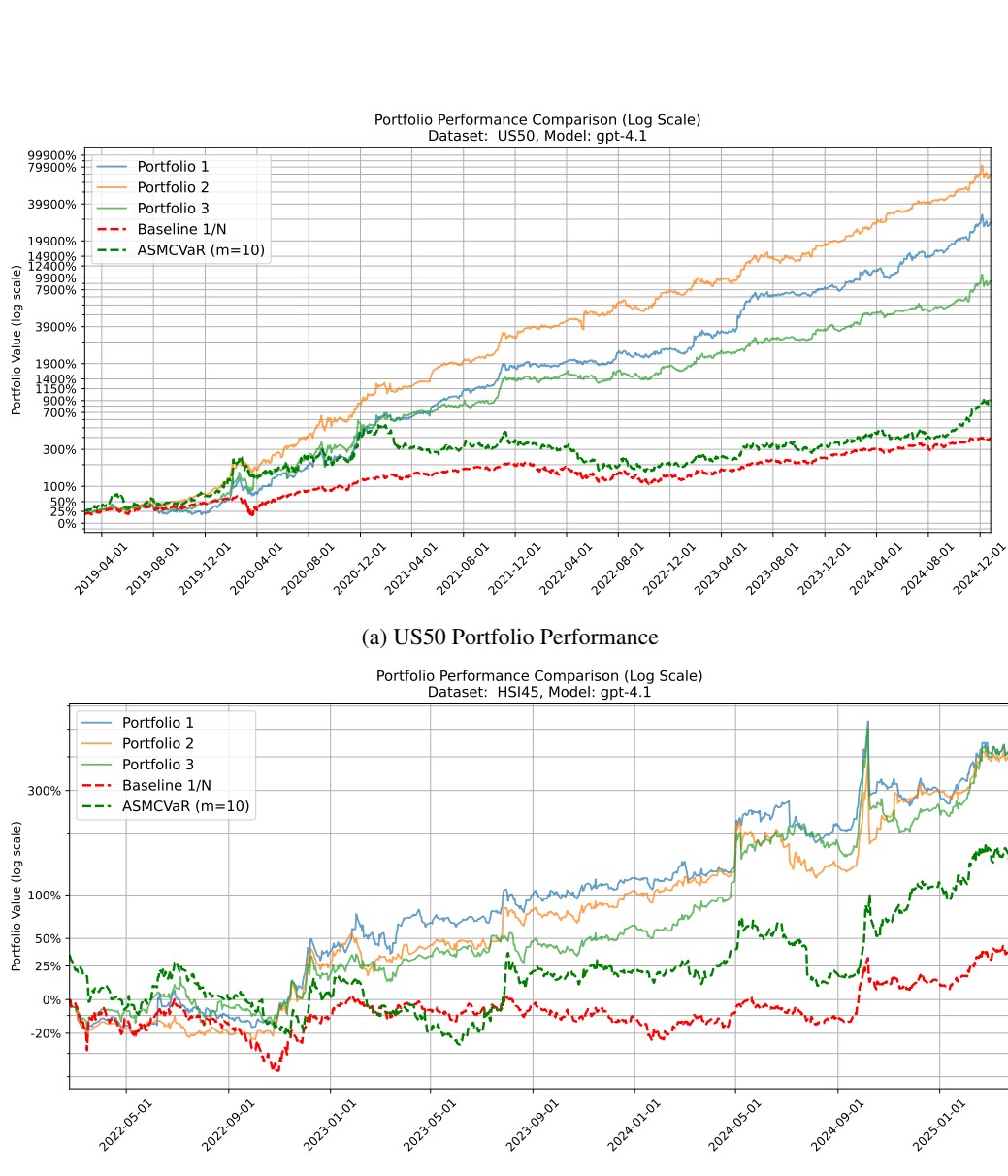

(a) US50 Portfolio Performance

(b) HSI45 Portfolio Performance

Figure 18: Factor-based sparse solution portfolio performance comparison. Both plots show cumulative returns on a logarithmic scale (y-axis), demonstrating the relative performance of different portfolio strategies. Figure 18a displays results for the US50, while Figure 18b shows performance for the HSI45.

