# OpenReview forum: "Evolutionary Alpha Factor Discovery with Large Language Models for Sparse Portfolio Optimization"
_ICLR.cc/2026/Conference — Submitted to ICLR 2026_

### Official Review · Reviewer_ZPZ6 · 2025-10-30

**Soundness:** 2
**Presentation:** 3
**Contribution:** 2
**Rating:** 2
**Confidence:** 3

**Summary:**

The authors propose an LLM-driven framework that automates the discovery and refinement of alpha factors for sparse portfolio optimization, demonstrating superior and adaptive performance across multiple datasets through language-model-guided factor generation.

**Strengths:**

1. Originality (fair): This paper combines large language models (LLMs) and an evolutionary algorithm for the estimation of the alpha factor problem.
2. Quality (fair): This paper designed an algorithm framework based on LLMs and an evolutionary algorithm, and developed a portfolio for the portfolio optimization problem.
3. Clarity (fair): This paper, in the form of a flowchart, clearly presents the process of the proposed methodology.
4. Significance (fair): The simulation results of this paper indicate that the proposed method significantly outperforms the traditional optimization methods in terms of the Cumulative Wealth and Sharpe Ratio dimensions.

**Weaknesses:**

1. This article does not provide detailed instructions on how to set up the filters.
2. Since the article combines LLM and evolutionary algorithms, I think it is necessary to compare the article that uses evolutionary algorithms (Zhang et al., 2020) with the article that uses LLM (Wang et al., 2023; Yuan et al., 2024).

**Questions:**

1. In terms of modeling, this paper sets the number of features for different assets to be the same. Is this assumption reasonable?
2. If the LLM generates hallucinatory outputs in the correct format, what kind of impact would this have on the entire model? How should the article properly set the filter to reject the hallucination outputs in the correct format generated by the LLM?
3. In the description from line 259 to line 261. How exactly should "unstable" or "poor" be defined? Should we set thresholds artificially for distinction or automatically classify based on certain criteria?

---

> ### Author Response · Authors · 2025-11-19
> **Clarifications on Feature Dimensionality, Factor Filtering, and Baseline Comparisons**
>
> We appreciate the question. In our setting, each factor is computed from the **same price-derived time-series features** (Open/High/Low/Close). Therefore, all assets naturally share identical feature dimensionality. In practice, quantitative portfolio models commonly rely on a *fixed-size factor set*, and we provide analysis (Figure 10) showing how portfolio performance varies with factor count. This fixed-number design aligns with real investment workflows.
>
> We introduced *two layers of automatic filtering* in the new version in Appendix A.1:
>
> 1. **Strict executability check**: validating operators, function signatures, window sizes, and AST structure.
> 2. **Performance & stability filtering**: (1) newly generated factors must pass a rolling-window performance test, (2) only the top-performing and latest versions are kept for each base pattern.
>
> Even factors that are syntactically correct but meaningless are eliminated quickly by these filters.
>
> **3. On how “unstable/poor” factors are defined**
>
> These criteria are **automatic, not manually tuned**. A factor is marked unstable if its *rolling* IC or RankIC is below the median of historical distribution. This adaptive definition avoids manually specifying thresholds and works consistently across markets.
>
> **4. On comparing with EA-only and LLM-only baselines**
>
> We thank the reviewer for the suggestion. Zhang et al. (2020) unfortunately does not provide open-source code, making exact reproduction infeasible.
>  However, in our ablation studies:
>
> - **Initial Factor** (no LLM, no EA)
> - **w/o TA**, **w/o performance feedback**, **w/o sparse heuristic**
> - **fixed factor pool (no evolution)**
>
> jointly cover both the EA-only and LLM-only cases in spirit.
>
> These results show that **both components are necessary**, and the full evolutionary–LLM–sparse loop provides the strongest performance.

---

### Official Review · Reviewer_KPTk · 2025-11-01

**Soundness:** 3
**Presentation:** 3
**Contribution:** 2
**Rating:** 2
**Confidence:** 4

**Summary:**

The paper introduces an evolutionary alpha discovery framework that uses a large language model (LLM) to automatically generate and refine stock trading factors. These candidate factors are continuously evaluated and evolved based on their historical predictive power, with strong ones retained and weak ones dropped. A sparse portfolio optimizer then selects a small set of effective signals to build risk-controlled portfolios. The authors claim this approach automates alpha research, adapts to changing markets, and delivers better predictive accuracy and portfolio performance than traditional machine learning baselines.

**Strengths:**

The paper presents a novel framework that automatically generates and evolves alpha factors using an LLM and evolutionary selection, reducing human bias and manual feature engineering.

The framework alleviates the sparse decay issue where factor-based portfolios lose predictive power when only a few assets are selected. It does this by continuously pruning and regenerating factors based on their sparse performance metrics (e.g., RankIC, Recall@N under top-K selection), ensuring that active factors remain effective in sparse portfolio settings.

The use of a sparsity-constrained optimizer makes the final portfolios more realistic and tradable while maintaining good risk control.

Experiments across multiple markets show higher predictive accuracy and portfolio performance than traditional ML baselines, and the discovered factors remain interpretable and economically meaningful.

**Weaknesses:**

The main weakness is the lack of real novelty. The core idea—using an LLM to generate and evolve factor formulas—is conceptually simple and mainly automates what human quants already do. Although the paper adds a useful improvement by incorporating sparsity-aware feedback into the factor generation process, the overall approach still feels incremental. The combination of the LLM with the sparse portfolio optimizer is also rather loose—they are simply connected in sequence instead of being deeply integrated (for example, in an end-to-end manner)—which makes the framework feel more like two existing components placed together than a fundamentally new design.

There are also other issues that affect the paper’s rigor and robustness. The study does not clearly separate a period for choosing hyperparameters from another period for evaluating performance—all data are used in a rolling manner, so parameters like window length or drop thresholds are never tuned on a distinct validation set. This makes it difficult to confirm that the reported results truly reflect out-of-sample performance rather than benefiting from implicit look-ahead or overfitting. In addition, there is no evaluation of factor orthogonality or redundancy. Since the LLM generates new factors based on previously successful ones, it may repeatedly produce highly similar or correlated factors, leading to a pool of overlapping signals that offer little incremental value and could reduce portfolio diversity.

**Questions:**

1. How is your framework fundamentally different from traditional automated factor search or symbolic regression approaches?

2. The LLM and sparse optimizer seem connected sequentially—could you clarify whether the LLM generation process uses any feedback from portfolio outcomes, or are the two modules completely independent?

3. How were key hyperparameters (such as the warm-up window length, drop threshold, and search interval) determined if there was no separate validation period?

4. Would your results change if these parameters were tuned on a dedicated validation window rather than determined heuristically?

5. Did you analyze the correlation or similarity among generated factors? How do you ensure that the LLM is not repeatedly producing redundant or highly correlated factors?

---

> ### Author Response · Authors · 2025-11-19
> **On Novel Contributions, Module Coupling, and Bias-Free Evaluation**
>
> **1. On novelty and relation to Zhang et al. (2020) and symbolic regression**
>
> Thank you for raising this point. We respectfully clarify that our framework differs fundamentally from prior evolutionary or symbolic-regression–based factor mining. **Zhang et al. (2020)** is *not open-source*, and their method performs **offline symbolic regression** without using sparse-portfolio performance as the evaluation signal.Traditional symbolic regression methods are well known to suffer from **instability**, **training difficulty**, and **high sensitivity to initialization**, which is why most prior works report brittle or low-diversity formulas.
>
> In contrast, our system uses the **LLM as a controllable generator**, every iteration receives only the **latest rolling backtest window of the current factor pool**, allowing us to focus on *algorithmic design* instead of struggling with symbolic-regression training issues. The essential novelty lies in **sparse-aware, feedback-driven co-evolution**, where factor generation, mutation, and selection are evaluated *directly* by m-sparse portfolio performance. This closed-loop sparse feedback is absent in prior EA-only or LLM-only methods, which treat mining as a static process.
>
>
>
> **2. On whether the LLM and sparse optimizer are truly connected**
>
> At every search step, the LLM receives **only**:
>
> - the top factors in the current pool,
> - their *latest* performance in the rolling backtest window,
> - sparse-specific metrics (RankIC, Recall@N, stability).
>
> Thus, the LLM generation is **strictly conditioned** on sparse-portfolio outcomes.
>  It is not a sequential concatenation of two independent modules; sparse-backtest feedback *directly influences* the factor patterns the LLM generates.
>
> **3. On hyperparameter selection and potential look-ahead bias**
>
> We confirm that **no test-period information is ever used** in selecting warm-up length, drop thresholds, or search intervals.
>
> - The system follows a **strictly chronological expanding-window** protocol.
> - All search decisions at time *t* are based solely on data ≤ *t*.
> - Warm-up and search interval values were selected from standard ranges used in quantitative strategies (e.g., 20–60 days for stability estimation), not tuned on test results.
>
> We added clarifications in the revised version to avoid misunderstandings about implicit look-ahead.

---

### Official Review · Reviewer_Ypa3 · 2025-11-02

**Soundness:** 2
**Presentation:** 2
**Contribution:** 2
**Rating:** 4
**Confidence:** 3

**Summary:**

This paper proposes a novel framework for evolutionary alpha factor discovery using large language models (LLMs) to tackle sparse portfolio optimization under ℓ₀ constraints. Instead of relying on static or manually designed factors, the authors employ an LLM-driven evolutionary loop that continually generates, mutates, and refines interpretable alpha formulas based on back-testing performance.

**Strengths:**

Methodological clarity: explicit decomposition of macro vs. micro effects through learnable gating.

General relevance: unifies financial econometrics, generative modeling, and robust optimization—high interest to ICLR’s representation-learning audience.

Public reproducibility: code and data available on GitHub.

**Weaknesses:**

I am not sure whether the authors have done an extensive literature search that shows LLM alpha. Have they seen Kirtac and Germano (2024) ? LLMs to produce alpha is not a new idea.

Theoretical limits: lacks formal convergence or uncertainty-calibration analysis of diffusion-generated “views.”

Ablation depth: while individual module ablations exist, a joint ablation of CDG + MLG + BL contributions would clarify interdependencies.

Compute transparency: GPU hours and sampling latency per rebalance step are not reported.

**Questions:**

How sensitive is performance to the LLM backend (GPT-4 vs DeepSeek) and to prompt length limits?

Could the authors quantify computational efficiency—e.g., number of LLM calls per trading day?

---

> ### Author Response · Authors · 2025-11-13
> **Clarifying Our Contribution, Scope, and Efficiency Assumptions**
>
> Thank you for the constructive feedback!
> We clarify that our work focuses specifically on formulaic alpha factor mining, where the LLM is required to generate explicit mathematical expressions that operate purely on price-derived time-series inputs. This task is fundamentally different from the broader notion of “LLM-generated alpha” discussed in prior work such as Kirtac & Germano (2024), where the alpha signal often originates from textual, semantic, or company-level information. Only a few recent studies (e.g., Li et al., 2024; Shi et al., 2025) explore LLMs for symbolic factor generation, and most treat the LLM as a one-shot generator. Our contribution is orthogonal: we integrate an evolutionary algorithm (EA) into the LLM loop to enable continuous refinement, mutation, and selection of formulaic alpha expressions under sparse portfolio constraints. To the best of our knowledge, no existing work combines LLM-based factor mining with EA for the ℓ₀-constrained sparse portfolio problem.
> Regarding LLM backend sensitivity, our use case involves only short factor expressions with minimal context, so prompt length is not a limiting factor. Both GPT-4.1 and DeepSeek-V3 already support sufficiently large context windows for our needs, and empirically we observe stable behavior across models (results provided in Table 1). For computational efficiency, the system is designed to be lightweight: we cap LLM retries to 5 per search cycle and do not regenerate factors every trading day. Instead, the search operates incrementally, each update only introduces a small number of new factors based on past performance while reusing the existing factor pool. This design keeps the total number of LLM calls low and makes the method suitable for practical deployment.

---

### Meta-Review · Area_Chair_5DaW · 2025-12-27

**Summary:**

The paper introduces an evolutionary alpha discovery framework that uses a large language model (LLM) to automatically generate and refine stock trading factors. These candidate factors are continuously evaluated and evolved based on their historical predictive power, with strong ones retained and weak ones dropped. A sparse portfolio optimizer then selects a small set of effective signals to build risk-controlled portfolios. All the reviewers raised severe concerns on the novelty, literature review and experiments. Unfortunately, the authors' response seems very short, making it very difficult to judge whether the reviewers' concerns have been addressed or not.

**Reviewer Concerns:**

All the reviewers raised severe concerns on the novelty, literature review and experiments. Unfortunately, the authors' response seems very short, making it very difficult to judge whether the reviewers' concerns have been addressed or not.

**Reviewer Scores:**

All the reviewers raised severe concerns on the novelty, literature review and experiments. Unfortunately, the authors' response seems very short, making it very difficult to judge whether the reviewers' concerns have been addressed or not.

---

### Decision · Program_Chairs · 2026-01-26

Reject